# Ush regulates hemocyte-specific gene expression, fatty acid metabolism and cell cycle progression and cooperates with dNuRD to orchestrate hematopoiesis

Jonathan Lenz[1], Robert Liefke[1,2], Julianne Funk[3], Samuel Shoup[1], Andrea Nist[4], Thorsten Stiewe[4], Robert Schulz[5†], Yumiko Tokusumi[5], Lea Albert[6], Hartmann Raifer[7], Klaus Förstemann[8], Olalla Vázquez[6], Tsuyoshi Tokusumi[5], Nancy Fossett[9], Alexander Brehm[1]*

**1** Institute of Molecular Biology and Tumor Research, Biomedical Research Center, Philipps-University, Marburg, Germany, **2** Department of Hematology, Oncology and Immunology, University Hospital Giessen and Marburg, Marburg, Germany, **3** Institute of Molecular Oncology, Philipps-University, Marburg, Germany, **4** Genomics Core Facility, Institute of Molecular Oncology, Member of the German Center for Lung Research (DZL), Philipps-University, Marburg, Germany, **5** Department of Biological Sciences, University of Notre Dame, Notre Dame, Indiana, United States of America, **6** Faculty of Chemistry, Philipps-University, Marburg, Germany, **7** Flow Cytometry Core Facility, Institute for Medical Microbiology and Hospital Hygiene, Biomedical Research Center, Philipps-University, Marburg, Germany, **8** Gene Center and Dept. of Biochemistry, Ludwig-Maximilians-Universität, München, Germany, **9** Center for Vascular and Inflammatory Diseases and the Department of Pathology, University of Maryland School of Medicine, Baltimore, Maryland, United States of America

† Deceased.
* brehm@imt.uni-marburg.de

**Data Availability Statement:** Raw and analysed data can be accessed at the GEO database: accession no. GSE146382. In particular this

## Abstract

The generation of lineage-specific gene expression programmes that alter proliferation capacity, metabolic profile and cell type-specific functions during differentiation from multi-potent stem cells to specialised cell types is crucial for development. During differentiation gene expression programmes are dynamically modulated by a complex interplay between sequence-specific transcription factors, associated cofactors and epigenetic regulators. Here, we study U-shaped (Ush), a multi-zinc finger protein that maintains the multipotency of stem cell-like hemocyte progenitors during *Drosophila* hematopoiesis. Using genome-wide approaches we reveal that Ush binds to promoters and enhancers and that it controls the expression of three gene classes that encode proteins relevant to stem cell-like functions and differentiation: cell cycle regulators, key metabolic enzymes and proteins conferring specific functions of differentiated hemocytes. We employ complementary biochemical approaches to characterise the molecular mechanisms of Ush-mediated gene regulation. We uncover distinct Ush isoforms one of which binds the Nucleosome Remodeling and Deacetylation (NuRD) complex using an evolutionary conserved peptide motif. Remarkably, the Ush/NuRD complex specifically contributes to the repression of lineage-specific genes but does not impact the expression of cell cycle regulators or metabolic genes. This reveals a mechanism that enables specific and concerted modulation of functionally related portions of a wider gene expression programme. Finally, we use genetic assays to demonstrate that

includes raw reads from RNA-seq experiments (3 replicates each: dsEGFP, dsUsh, dsMi-2; 4 replicates each: dsEGFP, dsUsh-B) and ChIP-seq experiments (1 replicate each: Ush-GFP input, Ush-GFP ChIP, dMi-2-GFP input, dMi-2-GFP ChIP). For RNA-seq the following analysed data are provided: Normalised counts (dsEGFP, dsUsh & dsMi-2 replicates), differentially expressed genes (dsEGFP vs. dsUsh; dsEGFP vs. dsMi-2), Normalised counts (dsEGFP & dsUsh-B replicates), differentially expressed genes (dsEGFP vs. dsUsh-B). Coverage tracks of Ush and dMi-2 ChIP-seq experiments as well as corresponding inputs are available for two different *Drosophila* genome versions (bigWig files for dm3 and dm6).

**Funding:** JL, RL, LA, OV and AB were funded by the Deutsche Forschungsgemeinschaft (DFG, German Research Foundation) - TRR 81/3 - 109546710. The funders had no role in study design, data collection and analysis, decision to publish, or preparation of the manuscript.

**Competing interests:** The authors have declared that no competing interests exist. Author Robert Schulz was unable to confirm their authorship contributions. On their behalf, the corresponding author has reported their contributions to the best of their knowledge.

Ush and NuRD regulate enhancer activity during hemocyte differentiation *in vivo* and that both cooperate to suppress the differentiation of lamellocytes, a highly specialised blood cell type. Our findings reveal that Ush coordinates proliferation, metabolism and cell type-specific activities by isoform-specific cooperation with an epigenetic regulator.

## Author summary

In multicellular organisms common progenitors differentiate into various kinds of specialised cells. During differentiation metabolic profiles and proliferation potentials are progressively adjusted and cell type-specific traits are established by the coordinated activation and inactivation of genes. Here we study U-shaped (Ush), a conserved gene regulator that acts during macrophage differentiation in *Drosophila melanogaster*. We uncover that Ush coordinates the activation and inactivation of three differentiation-related gene groups, thereby modulating lipid metabolism, promoting cell division and maintaining a progenitor state. These functions are conferred by different Ush protein isoforms and their associated co-factors. One such co-factor, the nucleosome remodeling and deacetylation complex dNuRD, contributes to progenitor state maintenance but is not required for other Ush-regulated processes. This exemplifies how a single gene regulator can simultaneously influence different aspects of cellular differentiation by employing protein isoforms and isoform-specific co-regulator interactions.

## Introduction

Establishment of gene expression programmes during differentiation involves a close cooperation between lineage-specific transcription factors and ubiquitously expressed epigenetic regulators. Transcription factors often possess sequence-specific DNA binding activities to target specific genes. There, they interact with epigenetic regulators, such as histone modifying enzymes or nucleosome remodelers, which alter chromatin structure. This facilitates the establishment and maintenance of appropriate levels of transcription. The molecular details of this interplay are complex and incompletely understood.

During hematopoiesis multipotent stem cells differentiate into diverse lineages to produce the many different blood cell types. Lineage-specific expression of RUNX1, PU.1 and GATA transcription factors play a prominent role in guiding these cell fate decisions [1]. These sequence-specific transcription factors cooperate with a host of cofactors and epigenetic regulators to establish lineage-appropriate gene expression programmes [2]. Many of the key regulators of hematopoiesis are conserved between vertebrates and invertebrates. *Drosophila* possesses a simple hematopoietic system that is composed of only three differentiated cell types [3]. The macrophage-like plasmatocytes make up the bulk of *Drosophila* hemocytes. The rarer crystal cells perform special roles in melanisation. Finally, the ultra-rare lamellocytes are only produced in significant numbers under extreme stress conditions. All three cell types can be derived from a common hemocyte precursor. Given its simplicity, *Drosophila* has proven to be an excellent, genetically tractable model to uncover fundamental principles of hematopoiesis.

Like its mammalian homolog FOG1, U-shaped (Ush) is a transcriptional cofactor that cooperates with GATA transcription factors to regulate key decisions during *Drosophila* hematopoiesis [4–8]. Ush and FOG1 do not bind DNA and are recruited to their sites of action by sequence-specific GATA transcription factors. Genetic studies support the view that Ush acts with the GATA transcription factor Serpent to maintain pluripotency of hemocyte progenitors

and suppress their differentiation [9–13]. Changes in Ush levels govern cell fate choice: The stem cell-like pro-hemocytes express high levels of Ush. Ush expression is downregulated to lower levels as pro-hemocytes differentiate into plasmatocytes and crystal cells and completely shut off during lamellocyte differentiation [10]. Previous analyses have identified a small number of Ush-regulated genes critical for the repression of hemocyte differentiation [14]. It is not known if Ush is dedicated to the regulation of these genes or if it controls more extensive transcriptional programmes. Moreover, the potential interplay between Ush and epigenetic regulators has not been studied.

Here, we use ChIP-seq and RNA-seq to determine genomewide Ush-occupied chromatin regions and Ush-regulated genes in the hemocyte-derived S2 cell line. Ush associates predominantly with promoters and enhancers at thousands of loci that are enriched for GATA binding sites. It regulates the expression of more than 1,800 genes which designates Ush as a major transcriptome regulator. Bioinformatic analyses uncover both activating as well as repressive functions of Ush. Ush uses these opposing activities to coordinately regulate distinct sets of genes: genes with hemocyte-related functions, genes that encodes key enzymes of fatty acid metabolism and genes coding for critical cell cycle regulators. These findings suggest that Ush does not only control the expression of hemocyte-specific genes, as implied by prior genetic studies, but that it also shapes the metabolic profile and maintains the proliferative potential of hemocytes. Indeed, prolonged depletion of Ush abrogates cell division and results in a pronounced G2/M block as detected by flowcytometric analysis.

Biochemically, we identify two major Ush isoforms. We use a variety of protein interaction assays to demonstrate that only the Ush-B isoform interacts with subunits of the Nucleosome Remodeling and Deacetylation (NuRD) complex *in vitro* and *in vivo*. Their interaction depends on a short N-terminal sequence specific for Ush-B. This sequence is related to the FOG repression motif with which FOG1 interacts with mammalian NuRD [15]. Thus, we have identified an evolutionary conserved, peptide based interaction mode between FOG1/Ush and NuRD. ChIP-seq highlights extensive colocalisation of Ush and the NuRD ATPase subunit dMi-2 on chromatin suggesting that the Ush/NuRD complex occupies thousands of regulatory sequences. RNA-seq analysis of the transcriptomes of dMi-2 and Ush-B-depleted cells identifies a common set of Ush-B/dMi-2 repressed genes with hemocyte-specific functions. By contrast, genes encoding enzymes involved in fatty acid metabolism and cell cycle regulation are not significantly affected by Ush-B/dMi-2. Accordingly, dMi-2 and Ush-B-depletion does not significantly affect the cell cycle profile of S2 cells. Thus, a specific Ush isoform and its specific interaction with an epigenetic regulator make a dedicated contribution to the regulation of only one of the three gene classes controlled by Ush.

Finally, we have used genetic loss-of-function approaches to define the roles of Ush and NuRD during hematopoiesis *in vivo*. We show that Ush as well as NuRD subunits are required for the restriction of enhancer activity in the lymph gland and that Ush and NuRD cooperate in the suppression of stress-induced lamellocytes.

Transcriptional factors make use of selective coregulators to establish and maintain cell lineage specific transcription programmes during mammalian hematopoiesis [2]. Our data substantially elaborates this paradigm by revealing alternative splicing and isoform-specific interactions as mechanisms to guide selective coregulator usage.

## Results

### Ush associates with promoters and enhancers

We used hemocyte-derived S2 cells as a model to define the molecular functions of Ush. Western blot analysis of whole cell extracts verified expression of Ush in S2 cells (**Fig 1A**, left panel).

An established Ush antibody reacted with several polypeptides (lane 1; [6]). The antibody signals for polypeptides with apparent molecular masses of 180 kDa and 220 kDa, respectively, were abrogated upon treatment of S2 cells with double stranded RNA directed against the 3' portion of the Ush mRNA (lane 2). This suggests that S2 cells express at least two different isoforms of Ush or that the protein is post-translationally modified. We employed a CRISPR approach to insert GFP- or FLAG-tag coding sequences at the 3' end of the Ush gene (**S1A and S1B Fig**). Western blot analysis of nuclear extracts from these cell lines using GFP or FLAG antibody likewise detected two major polypeptides (**Fig 1A**, right panel).

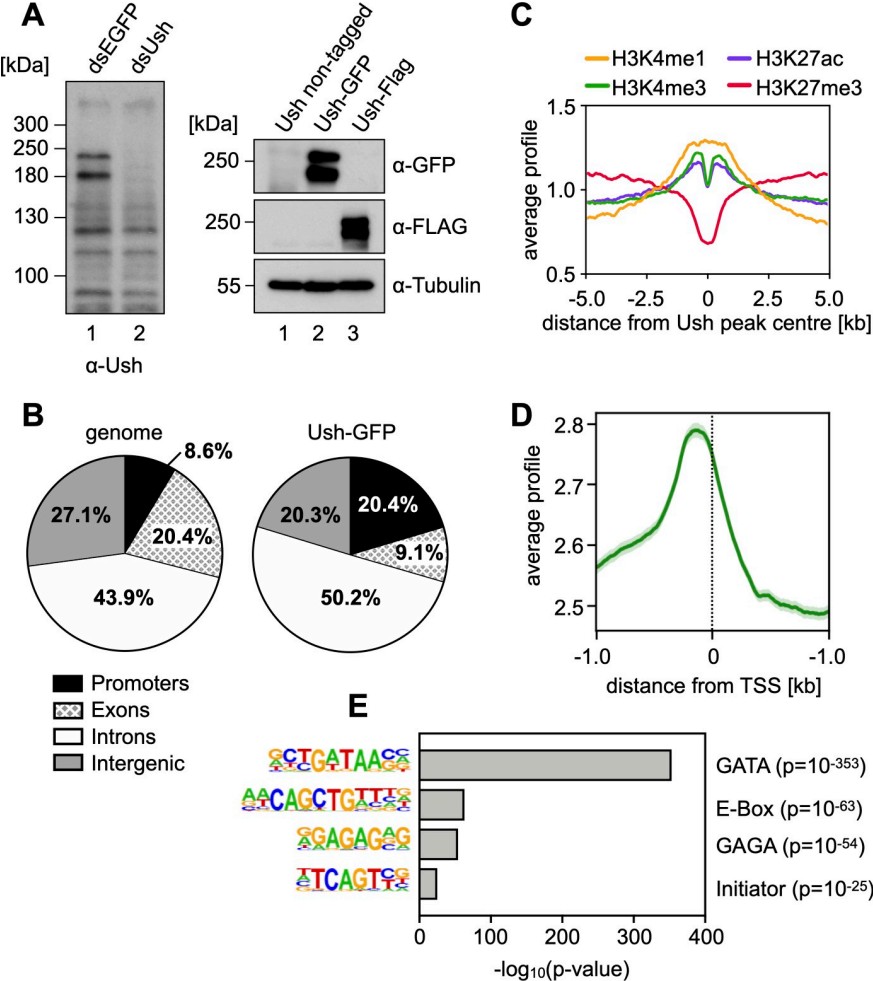

**Fig 1. Ush is expressed in S2 cells and binds to regulatory elements. A** Left panel: S2 cells were transfected with control dsRNA (dsEGFP) or dsRNA against Ush (dsUsh) and harvested after four days. Whole cell lysates were probed on Western blot using an antibody against Ush. Right panel: A GFP- or FLAG-tag sequence was inserted at the 3' end of the Ush gene in S2 cells using CRISPR/Cas9-mediated genome editing. Nuclear extracts of control cells and cells expressing Ush-GFP or Ush-FLAG was probed on Western blot using a GFP or FLAG antibody. Tubulin signal serves as loading control. **B** Genomic distribution of Ush-GFP binding sites identified by anti-GFP ChIP-seq. Fraction of Ush peaks found in each genomic location are shown in the right chart. Fractions of genomic locations in the *Drosophila* genome serve as reference (left chart). **C** Distribution of histone modifications surrounding Ush-bound regions. Signals of H3K4me1 (yellow), H3K4me3 (green), H3K27ac (blue) and H3K27me3 (red) are displayed within a region of 10 kb surrounding Ush peaks. **D** Distribution of Ush occupancy at transcription start sites (TSS). Average Ush binding (green) was evaluated in a 2 kb region surrounding all genomic TSS. Standard error is depicted in light green. **E** Analysis of DNA sequence motifs enriched at Ush binding sites. The enriched motif is depicted on the left and the corresponding transcription factor on the right. The -log$_{10}$(p-value) for the enrichment of each motif is plotted and p-values are indicated on the right.

We next determined the genomewide chromatin binding pattern of Ush by anti-GFP chromatin immunopecipitation followed by high throughput sequencing (ChIP-seq). This identified 7012 genomic regions bound by Ush-GFP. Ush occupied sites were strongly enriched in promoters and moderately enriched in introns, which in the *Drosophila* genome often harbour enhancers (**Fig 1B**). Ush-bound regions were positively correlated with higher levels of H3K4 monomethylation (H3K4me1), H3K4 trimethylation (H3K4me3) and H3K27 acetylation (H3K27ac) - three histone modifications that are characteristic for active promoters (H3K4me3 and H3K27ac) and enhancers (H3K4me1 and H3K27ac) (**Fig 1C**). By contrast, Ush-occupied sites were on average depleted of H3K27 trimethylated (H3K27me3) nucleosomes, which are predominantly associated with genes that are stably silenced by Polycomb complexes PRC1 and PRC2. Concordant with histone modification patterns, elevated Ush levels were found directly upstream of transcriptional start sites (TSS), suggesting that Ush occupies gene promoter sequences (**Fig 1D**). A motif analysis revealed that Ush bound regions are in fact enriched for transcription factor binding sites, including GATA-, E-box-, GAGA- and Initiator motifs (**Fig 1E**). Of these the GATA motif was by far the most strongly enriched motif consistent with the established genetic and physical interactions between Ush and GATA transcription factors [4–8].

Collectively, these results suggest that Ush preferentially occupies gene regulatory sequences such as promoters and enhancers and that it is predominantly associated with transcription factors such as GATA factors. However, our findings also hint towards a possible complex formation with bHLH transcription factors, GAGA factor and general transcription factors binding to the initiator element. Given the high number of Ush bound genes we hypothesised that Ush plays a significant role in regulating the S2 transcriptome.

## Ush is a major regulator of transcription

We depleted Ush from S2 cells by RNAi using a double stranded RNA that targets all Ush isoforms (**Fig 1A**). We then performed RNA-seq to analyse the resulting transcriptome changes. The levels of 1828 transcripts were significantly changed in Ush-depleted cells (adj. p < 0.01) supporting the hypothesis that Ush is a major transcriptional regulator. The majority of these transcripts (1268) was upregulated following Ush depletion suggesting that Ush predominantly represses transcription (**Fig 2A**). Nevertheless, a significant number (560) of differentially expressed genes were downregulated in Ush depleted cells indicating that Ush can also activate or maintain higher levels of transcription.

Comparison of the RNA-seq and ChIP-seq datasets revealed that approximately half of Ush-repressed genes (651 of 1268, 51%) and one third of Ush-activated genes (175 of 560, 31%) contain a Ush ChIP-seq peak in the promoter or gene body (**Fig 2B**). These 826 genes are, therefore, likely to be direct transcriptional targets of Ush.

## Ush regulates genes with hemocyte, metabolic and cell cycle functions

A gene ontology analysis of Ush regulated genes revealed strong enrichment of three main classes of genes: (1) genes involved in hemocyte functions (139 genes), (2) genes involved in lipid and fatty acid metabolism (199 genes) and (3) genes involved in the cell cycle (176 genes) (**Fig 2C** and **2D**).

Our finding that Ush regulates genes involved in hemocyte functions agrees well with previous genetic work: Ush has long been established as a dosage-dependent repressor of hemocyte differentiation in *Drosophila* [10]. In the embryo Ush antagonises the expression of the transcription factor Lozenge (Lz) which is essential for crystal cell differentiation [5,6,16]. Crystal cell differentiation is accompanied by reduced Ush expression and consequent derepression of

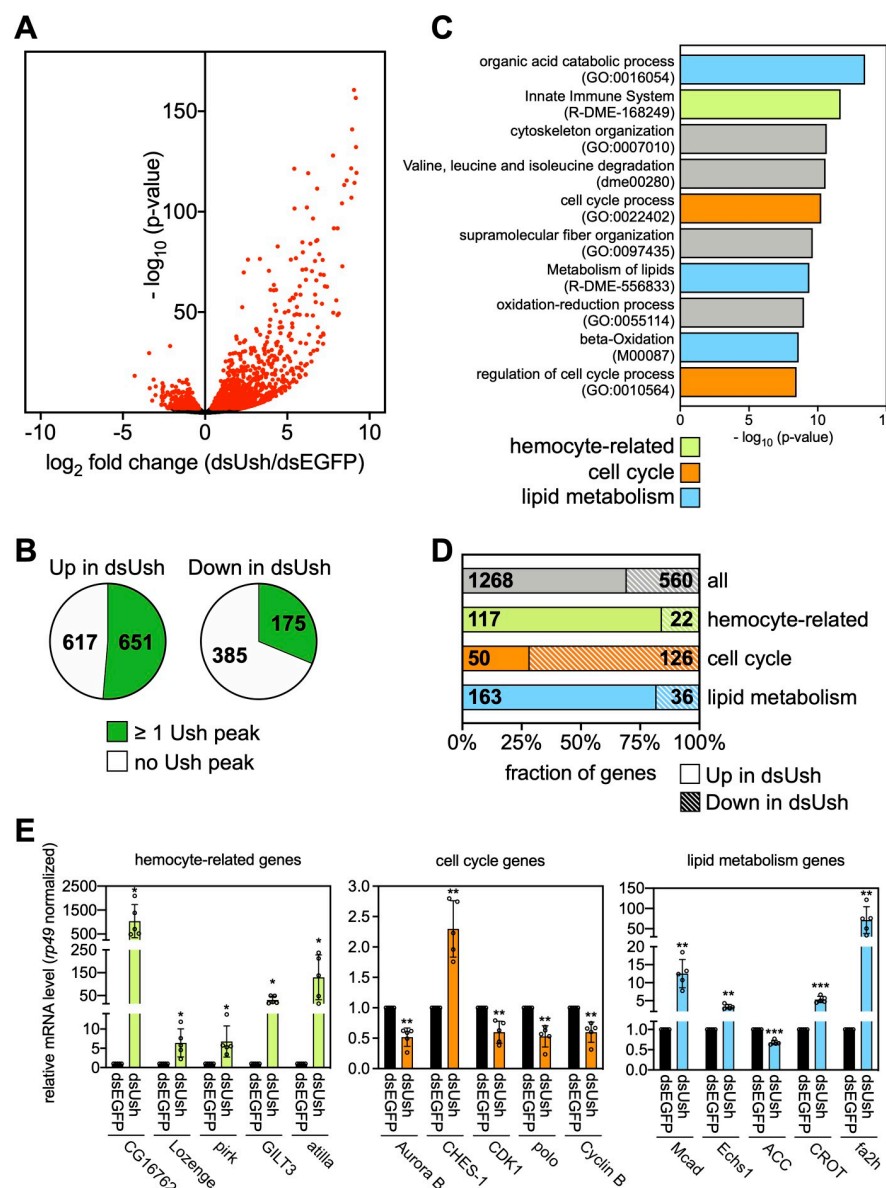

**Fig 2. Ush regulates the S2 cell transcriptome. A** Volcano plot of deregulated genes upon depletion of Ush in S2 cells. The -$\log_{10}$(p-value) is plotted against the $\log_2$ fold change of counts per gene in Ush-depleted (dsUsh) vs. control cells (dsEGFP). Red dots represent significantly deregulated genes (adj. p < 0.01) obtained from biological triplicates (n = 3). **B** Enrichment of Ush at Ush-regulated genes. Fraction of genes repressed (left chart) or activated by Ush (right chart) that contain at least one Ush peak is indicated in green. **C** Gene ontology analysis of Ush-regulated genes. GO terms associated with lipid metabolism (blue), hemocyte-specific functions (green) and cell cycle (orange) are highlighted respectively. **D** Genes contributing to the three GO term classes were divided into a Ush-activated (shaded) and a Ush-repressed (solid) fraction. The entirety of all Ush-regulated genes serves as reference (grey). Gene numbers in each fraction are indicated. **E** Representative genes from all three gene classes were analysed upon depletion of Ush (dsUsh) by RT-qPCR. Gene names are indicated below. Expression was calculated relative to control treated cells (dsEGFP) and normalised using the mRNA levels of *rp49*. Error bars represent the standard deviation from biological replicates (n = 5) (T-test: *** p < 0.001, ** p < 0.01, * p < 0.05). Individual values of each replicate are displayed as circles.

*lz*. Indeed, we find that the *lz* gene is bound by Ush and derepressed following Ush depletion suggesting that it is a direct target of Ush (**Figs 2E and S2**). This demonstrates that genetic relationships identified in fly embryos are recapitulated in S2 cells.

Unexpectedly, Ush also regulates a large number of genes which are involved in lipid metabolism and cell cycle control. Notably, cell cycle genes were mostly dependent on Ush for their robust expression while hemocyte- and metabolism-related genes were mostly repressed by Ush (**Fig 2D**). This suggests that the repressing and activating activities of Ush are predominantly used to control distinct transcription programmes that are modulating different cellular outcomes including hemocyte-specific functions, metabolic profile and cell cycle progression.

We selected representative genes from each class to confirm their regulation by Ush by RT-qPCR following Ush depletion (**Fig 2E**). Some of these genes had Ush ChIP-seq peaks within gene body and/or promoter and are, therefore, putative direct targets (CG16267, pirk, GILT3, Lozenge, CHES-1, Cyclin B, Mcad, Echs1, ACC, fa2h). Others were not bound by Ush and represent genes that might be indirectly regulated by Ush (Attila, AurB, CDK1, polo, CROT) (S1 Table). The levels of all five mRNAs encoding genes with hemocyte-related functions were increased by factors between five fold and about one thousand fold. Four cell cycle genes were downregulated upon Ush knockdown. These include important positive regulators of mitosis such as CDK1, polo, Cyclin B and Aurora B. By contrast, the forkhead transcription factor CHES-1, which in mammals has anti-proliferative activity, showed increased RNA expression [17]. Mcad (an acyl CoA dehydrogenase), Echs1 (Enoyl coenzyme A hydrolase), fa2h (fatty acid 2-hydroxylase) and CROT (a carnitin acyl transferase) collaborate in the degradation of fatty acids and the production of NADH, FADH2 and acetyl CoA. The levels of RNAs encoding these enzymes all increase upon Ush depletion. By contrast, levels of the RNA encoding acetyl CoA carboxylase (ACC), a key enzyme of fatty acid synthesis, decrease.

Thus, Ush appears to regulate different cellular processes in a coordinated fashion. It increases the expression of genes required for progression through mitosis and decreases the expression of an anti-proliferative gene. Likewise, it favours the expression of enzymes essential for fatty acid degradation while simultaneously lowering the expression of an enzyme that catalyses a key step in fatty acid synthesis.

## Ush is essential for cell cycle progression

A prediction from these observations is that proliferation should be adversely affected in Ush depleted cells. We simultaneously depleted all Ush isoforms by RNAi using two alternative double stranded RNAs. Compared to control cells that were treated with double stranded RNA targeting luciferase, Ush depletion dramatically decreased proliferation (**Fig 3A**). These cells were still viable which suggests that the observed reduction in cell number was not due to cell death (**Fig 3B**). We subjected S2 cells to flow cytometry after PI staining of DNA to determine the cell cycle profiles of control cells and Ush-depleted cells. Compared to control cells, Ush-depleted cells showed a pronounced reduction of cells with a 2n DNA complement and an accumulation of cells with a 4n complement. This suggests that Ush-depleted cells can replicate their genome but fail to enter or proceed through mitosis (**Fig 3C and 3D**). We then asked if this apparent G2/M block was accompanied by changes in the levels of mitotic cyclins which are required for progression into M phase. Again, we used two independent double stranded RNAs to deplete all Ush isoforms and determined Cyclin A and Cyclin B protein levels by Western blot (**Fig 3E**). Protein concentrations of both cyclins were reduced in Ush-depleted cells (compare controls in lanes 1 and 2 with lanes 3 and 4). Given that Ush depletion also results in a decrease of Cyclin B mRNA levels (**Fig 2E**) these data indicate that Ush promotes progression through the cell cycle, at least in part, by supporting the transcription of mitotic cyclins.

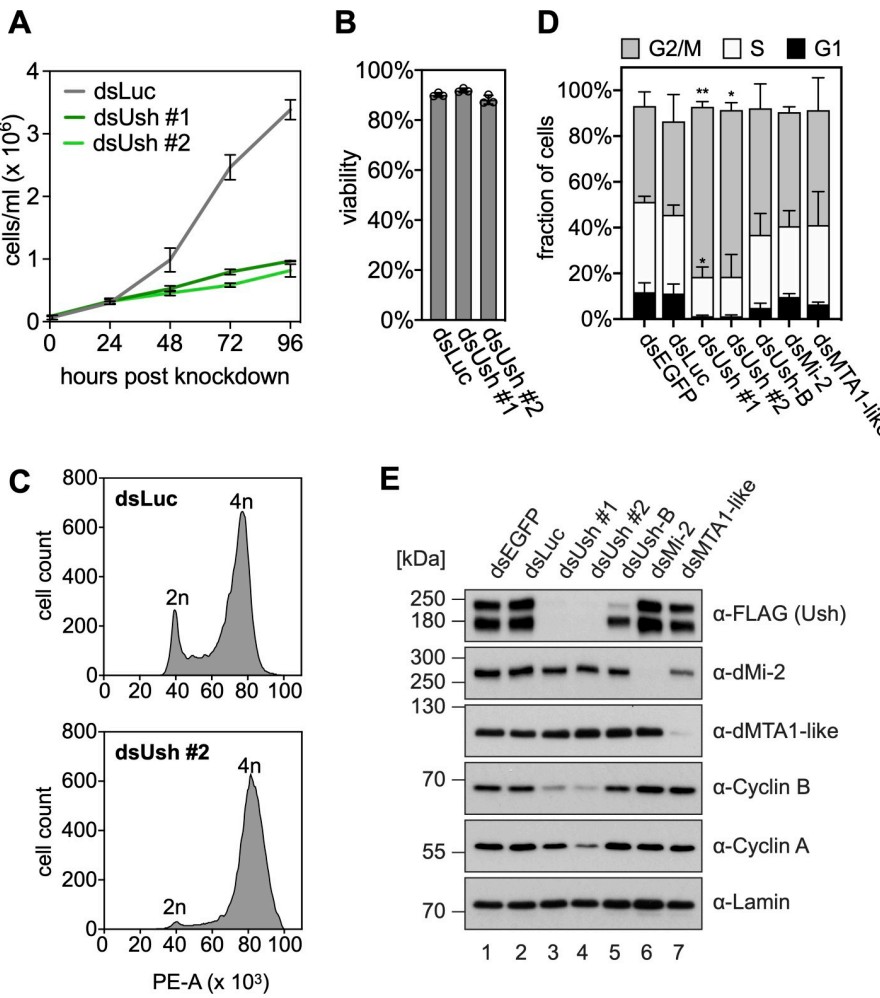

**Fig 3. Ush is necessary for cell cycle progression. A** Proliferation assay of S2 cells upon depletion of Ush. Cells were transfected with control dsRNA (dsLuc, grey) or with two different dsRNA constructs directed against Ush (dsUsh #1 in dark green, dsUsh #2 in light green). Cell numbers were determined every 24 hours. Error bars represent the standard deviation from biological triplicates (n = 3). **B** Viability assay of S2 cells upon depletion of Ush. Viability of cells transfected with control dsRNA (dsLuc) or dsRNA constructs targeting Ush (dsUsh #1 and dsUsh #2) was measured 96 hours post transfection. Error bars represent the standard deviation from biological triplicates (n = 3) and individual values are indicated with circles. **C** Flow cytometry following PI-staining of Ush-depleted (dsUsh #2) and control S2 cells (dsLuc). dsRNA-transfected cells were fixed, stained with PI and subjected to flow cytometry. Histograms show the number of cells plotted against the PI signal (Area of PE channel). The diploid cell population (2n) and cells that have undergone replication (4n) are indicated. **D** Quantification of cell populations obtained from flow cytometry of PI-stained cells upon depletion of indicated proteins (G1 phase: black, S phase: white, G2/M phase: grey). Error bars represent the standard deviation from biological triplicates (n = 3) (T-test: ** $p < 0.01$, * $p < 0.05$). **E** Western blot of whole cell extracts from S2 cells expressing endogenously FLAG-tagged Ush upon depletion of the proteins indicated above. Antibodies used for detection are indicated on the right. Lamin signal serves as loading control.

## Ush isoforms

How can Ush support the transcription of cell cycle genes and at the same time repress the transcription of many lipid metabolism- and hemocyte-related genes? We considered the possibilities that the activating and repressing functions of Ush are mediated by different Ush isoforms and/or association with different cofactors.

Indeed, the Ush gene structure predicts the expression of at least five different mRNAs generated by usage of alternative promoters and by alternative splicing (**Figs 4A** and **S3**). These mRNAs encode three Ush proteins that share a 1175 amino acids region at their C-termini which encompasses nine zinc fingers (**Fig 4B**). The three Ush isoforms differ in their unique short N-termini. Isoform Ush-D gives rise to a 1175 amino acid protein. Isoforms Ush-A and Ush-C produce two identical proteins which possess an additional 16 amino acid N-terminal extension (from hereon referred to as Ush-A) that is not present in Ush-D. Ush-B and Ush-E generate two identical proteins with a 23 amino acid N-terminal extension (from hereon referred to as Ush-B). As illustrated in **Fig 4B**, the first 7 and 14 amino acids of Ush-A and Ush-B, respectively, are unique to these isoforms. Our transcriptome data demonstrates expression of exons encoding both of these unique N-termini providing support for the expression of at least two different Ush protein isoforms in S2 cells (**S3 Fig**). If Ush isoforms do indeed possess isoform-specific functions they are likely to be mediated by these short N-terminal sequences.

An unbiased, large scale proteomic screen has previously identified several candidate interactors of Ush in S2 cells [18]. These include 6 subunits of the dNuRD complex. We immunoprecipitated nuclear extracts from S2 cells expressing FLAG-tagged Ush to verify these interactions (**Figs 1A and 4C**). Western blot analysis of the immunoprecipitate demonstrated that several subunits of the dNuRD complex coprecipitate with Ush. Importantly, dMEP-1, the signature subunit of the dMi-2-containing dMec complex, was not recovered [19]. This suggests that Ush specifically associates with the dNuRD complex but not with the dMec complex.

We also asked if immunoprecipitation of dMi-2 would coprecipitate Ush. Again, we used a CRISPR approach to add a FLAG-tag to the C-terminus of endogenous dMi-2 (**S1C–S1E Fig**). Western blot analysis of anti-FLAG immunoprecipitates from nuclear extract of these cells revealed that only the slower migrating of the two major Ush polypeptides coprecipitated with dMi-2 (**Fig 4D**, compare lanes 2 and 5). We used Ush isoform-specific RNA interference to identify Ush isoforms. The slower migrating isoform was efficiently depleted when cells were treated with double stranded RNA targeting an RNA region specific for Ush-B (lane 3). Immunoprecipitation of dMi-2-FLAG from nuclear extracts of Ush-B depleted cells failed to coprecipitate Ush protein (lane 6). We conclude that dMi-2 specifically forms a complex with the Ush-B isoform.

Inspection of the unique N-terminal sequence of Ush-B revealed that the first 9 amino acids are identical to the FOG repression motif (**Fig 4E**). This motif mediates interaction between several zinc finger transcription factors, including FOG1, and NuRD in mammalian cells [15,20–24]. We hypothesised that this motif does also mediate the interaction between Ush-B and dNuRD and that such a peptide-based NuRD binding mechanism is conserved between mammals and *Drosophila*. To test this hypothesis we incubated a GST fusion containing the N-terminus of mouse FOG1 (amino acids 1-45) with nuclear extracts of *Drosophila* S2 cells and *Drosophila* embryos (**Fig 4F**). All five dNuRD subunits we assayed interacted with GST-FOG1 but not with the GST control. We did not detect binding of the dMec subunit dMEP-1, the dMi-2 paralogue dCHD3 which does not assemble into a dNuRD complex and several components of other repressive chromatin regulating complexes (dPc, dE(z), dLSD1).

In order to compare the affinity of dNuRD for binding the FOG1 and Ush N-termini we designed 15 amino acid peptides derived from the FOG1 and the Ush-B N-termini (FOG1-wt, Ush-wt; **Fig 4G**). In addition, we generated mutant versions of these peptides where three amino acids important for binding of mammalian NuRD to FOG1 where changed (FOG1--mut, Ush-mut) [15]. As an additional control we used FOG1 and Ush-B peptides with scrambled sequences. We then competed binding of dNuRD to the GST-FOG1 fusion with these

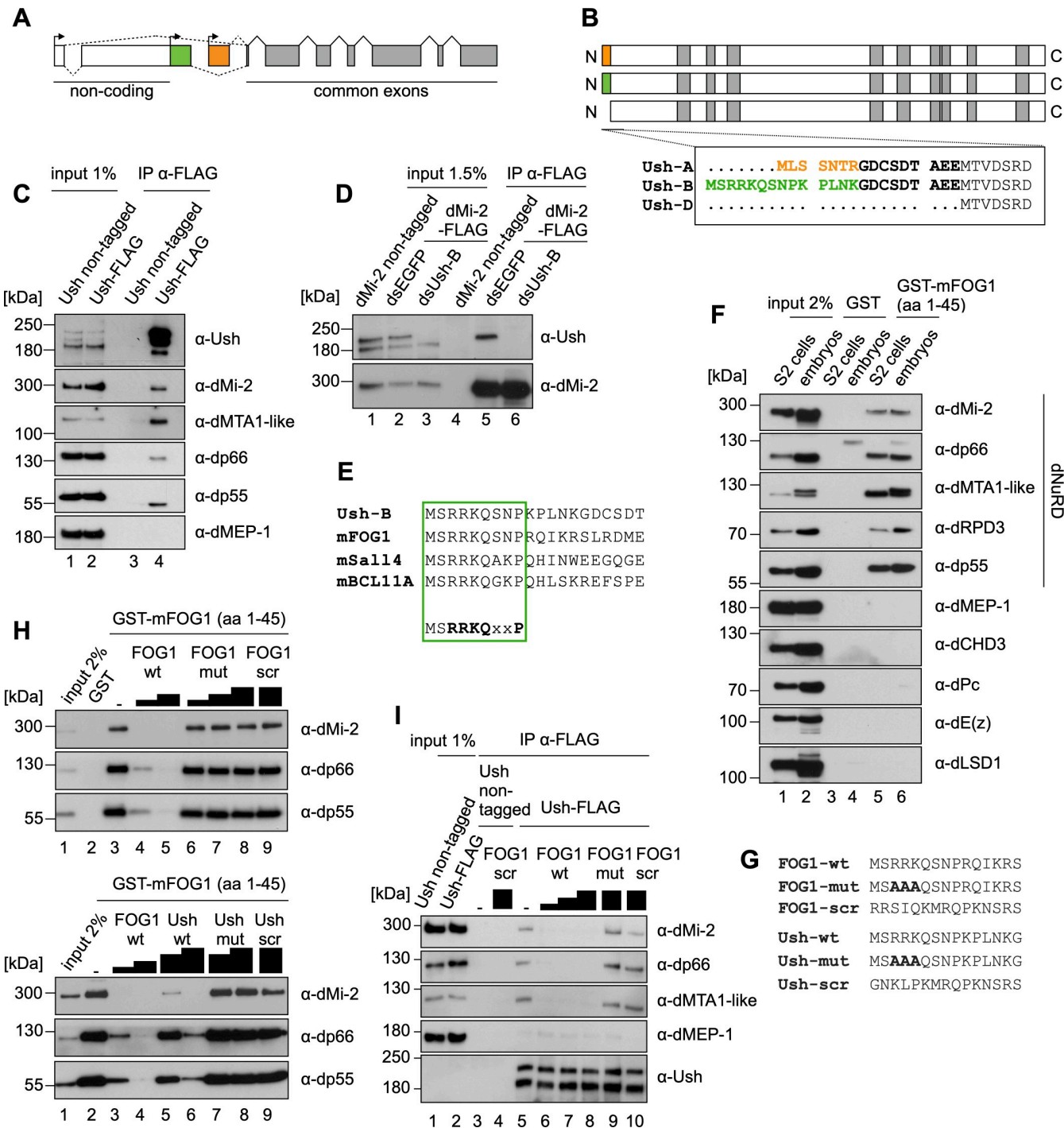

**Fig 4. The Ush isoform Ush-B interacts with NuRD via a conserved N-terminal motif. A** Schematic structure of the Ush gene locus. Boxes represent exons and the connecting lines indicate splicing events (dashed lines: alternative splicing). Exons marked in grey are common to all Ush isoforms. (Sections of) exons marked in white are untranslated. Possible transcriptional start sites are indicated by arrows. **B** Scheme of polypeptides generated from the Ush gene. Zinc finger domains are marked in grey. Sequences in the box indicate the N-termini of Ush proteins emanating from five possible Ush mRNAs. Isoform specific N-termini are marked in orange and green. **C** Anti-FLAG immunoprecipitation of nuclear extract from control and Ush-FLAG expressing cells. Antibodies used for examination of co-precipitation by Western blot are indicated on the right. **D** Anti-FLAG immunoprecipitation of nuclear extract from control and dMi-2-FLAG expressing cells following Ush-B depletion (dsUsh-B) or cells transfected with control dsRNA (dsEGFP). Co-precipitation of Ush was determined by Western blot. **E** Sequence alignment of the Ush-B N-terminus with N-terminal sequences from murine proteins containing the FOG repression motif (in bolt letters below). **F** GST pulldown from nuclear extracts of S2 cells or *Drosophila* embryos using the first 45 amino acids of murine FOG1 fused to GST (GST-mFOG1 (aa 1-45)) or control bait (GST). Interacting proteins were analysed by Western blot against NuRD complex components

(specified on the right) and additional chromatin-regulating proteins. Antibodies used for immuno-detection are indicated on the right. **G** Sequences of peptides derived from FOG1 and Ush-B N-termini that were used in competition experiments. **H** GST pulldown assays from S2 cell nuclear extracts using the GST-mFOG1(1-45) fusion protein. Pulldown reactions were performed in presence of different concentrations of the indicated peptides (FOG1 derived peptides: top panel; Ush-B derived peptides: bottom panel). Interaction of NuRD with the GST-fusion was detected by Western blot using antibodies indicated on the right. **I** Anti-FLAG immunoprecipitation of nuclear extract from control and Ush-FLAG expressing cells in presence of FOG1 derived peptides. The identity of peptides and the amount used is indicated above. Antibodies used for examination of co-precipitation by Western blot are indicated on the right.

peptides (**Fig 4H**). Both the FOG1-wt and Ush-wt peptides efficiently abrogated binding of the dNuRD subunits dMi-2, dp66 and dp55 to GST-FOG1. A higher excess of Ush-wt peptide was required for complete inhibition of binding suggesting that the affinity of the FOG1-wt peptide for binding to dNuRD is higher than the affinity of the Ush-wt peptide under our experimental conditions. Importantly, the mutant versions of both peptides as well as the scrambled controls did not compete for binding.

We next sought to test if FOG repression motif containing peptides are able to disrupt dNuRD/Ush complexes that have formed *in vivo*. We carried out FLAG-immunoprecipitation from nuclear extracts of S2 cells expressing FLAG-tagged Ush in the absence or presence of FOG1-wt, FOG1-mut or scrambled peptides and then analysed the immunoprecipitates by Western blot (**Fig 4I**). The FOG1-wt but not the FOG1-mut or the scrambled peptides disrupted the dNuRD/Ush complex.

These results suggest that the FOG repression motif present in the N-terminus of Ush is critical for binding dNuRD. Moreover, residues within the FOG repression motif that are essential for binding mammalian NuRD complexes are also critical for contacting the *Drosophila* NuRD complex. Taken together our analysis has revealed a highly conserved, peptide-based mechanism that mediates an isoform-specific interaction between Ush and dNuRD.

## dMi-2 and Ush co-occupy many sites on chromatin

We asked if Ush and dNuRD do not only interact in solution but are also associated on chromatin. We determined the genomewide chromatin binding of dMi-2-GFP by ChIP-seq. This identified 8459 peaks. Comparison of this dataset with two dMi-2 ChIP-seq profiles generated previously using two different dMi-2 antibodies demonstrated a highly similar binding pattern between the datasets (**S4 Fig**).

Comparison of our Ush-GFP and dMi-2-GFP ChIP-seq datasets uncovered a remarkable degree of co-localisation of the two proteins. About two thirds (64.9%) of Ush peaks overlapped with dMi-2 peaks (**Fig 5A**). Moreover, regions with strong Ush binding generally also displayed elevated dMi-2 binding (**Fig 5B**). Visual inspection of Ush and dMi-2 ChIP-seq profiles confirmed co-occupancy at many promoters, introns and intergenic regions (**Fig 5C**) while also revealing regions that are exclusively occupied by only one of the two factors (**Fig 5C, first panel**). We then assigned Ush/dMi-2 co-occupied regions as well as "Ush-only" and "dMi-2-only" ChIP-seq peaks to genomic regions (**Fig 5D**). Ush-only peaks show a strong preference of introns. dMi-2-only peaks on the other hand are most strongly enriched at promoters. Co-occupied regions show preferential association with both promoters and introns. In agreement with these findings analysis of histone marks at Ush-only peaks revealed a strong enrichment of H3K4me1, a histone modification that is characteristic for enhancers (**Fig 5E**). dMi-2-only peaks contained high levels of H3K4me3, a hallmark of active promoters. Co-occupied regions displayed elevated levels of both H3K4me1 and H3K4me3.

Collectively, these results supports the hypothesis that Ush/NuRD complexes act at regulatory regions such as promoters and enhancers.

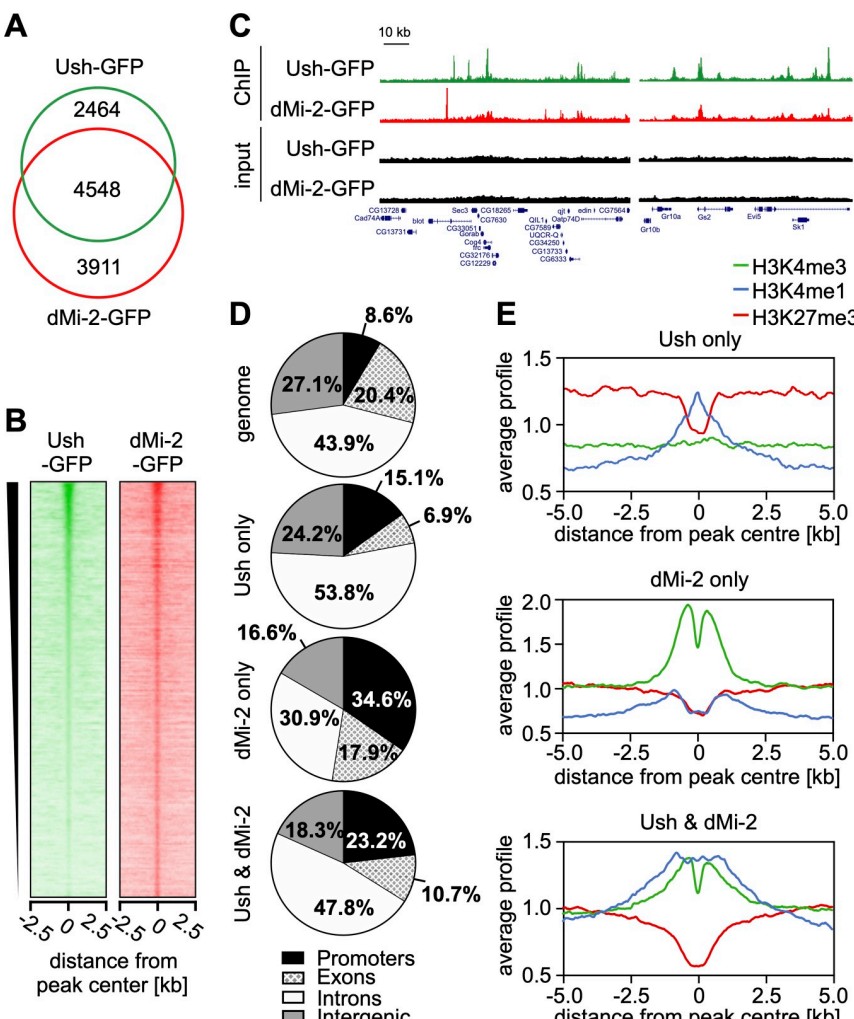

**Fig 5. Ush and dMi-2 co-localise on chromatin. A** Venn diagram of loci bound by Ush-GFP (green) and dMi-2-GFP (red) determined by anti-GFP ChIP-seq. Numbers of peaks are indicated in each section. **B** Heatmap of Ush-GFP and dMi-2-GFP signals centred at Ush-bound regions and sorted by Ush signal intensity. A region of 5 kb surrounding the Ush peak is displayed. **C** Genome browser snapshots of exemplary regions displaying Ush (green) and dMi-2 (red) occupancy. Input signals are shown in black. Location of genes is displayed below with boxes indicating exons. Scale bar represents a distance of 10 kb. **D** Genomic distribution of regions identified by anti-GFP ChIP-seq that were bound by Ush only, dMi-2 only or by both Ush and dMi-2 (indicated on the left). Fraction of peaks found in each genomic location are displayed. Fractions of genomic locations in the *Drosophila* genome serve as reference (top chart). **E** Distribution of histone modifications surrounding regions bound by Ush only (top), dMi-2 only (middle) or both Ush and dMi-2 (bottom). Signals of H3K4me3 (green), H3K4me1 (blue) and H3K27me3 (red) are displayed within a region of 10 kb surrounding peak centres.

Taken together these results suggest that Ush and dNuRD are indeed associated on chromatin. The Ush/dNuRD complex binds regulatory sequences indicating that Ush-B and dNuRD might cooperate in the regulation of transcription.

## Ush-B and dMi-2 regulate hemocyte-related genes

We have identified three classes of genes that display significant expression changes when all Ush isoforms are depleted simultaneously: genes related to hemocyte functions, genes encoding enzymes of the lipid metabolism and genes involved in cell cycle progression. We sought

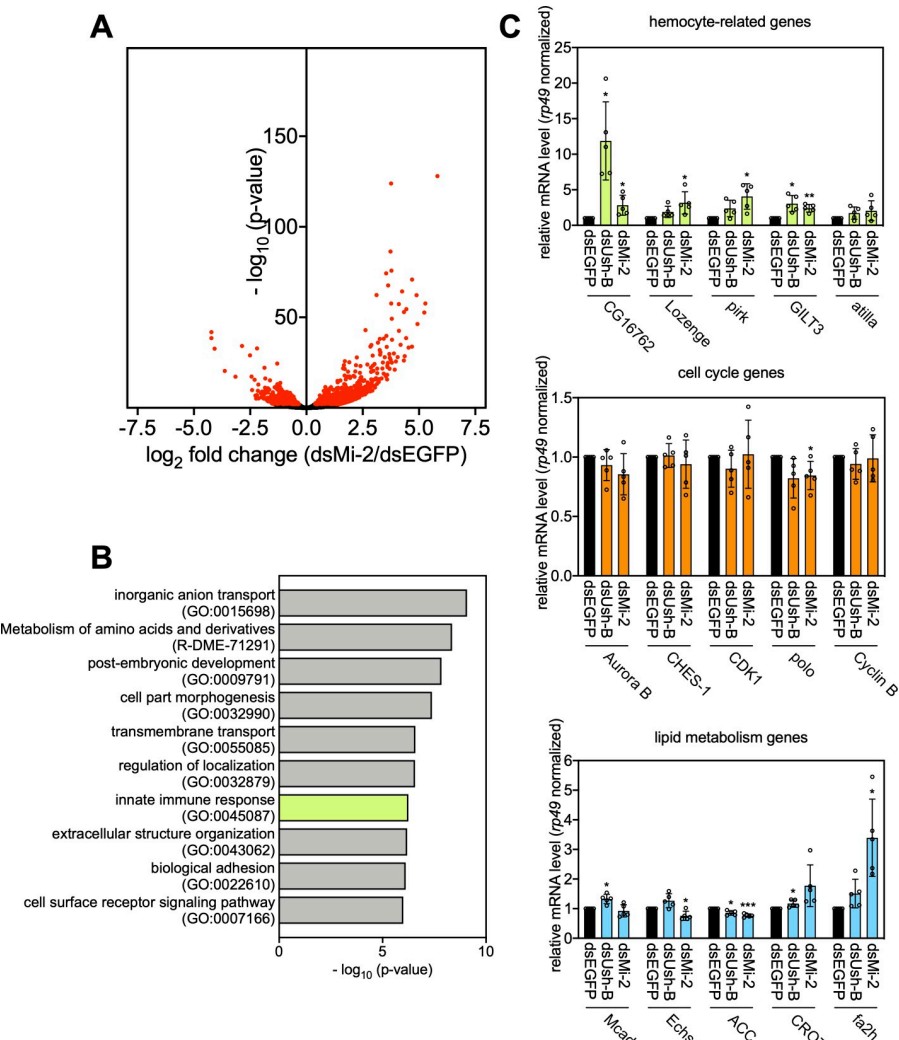

**Fig 6. dMi-2 and Ush-B regulate genes associated with hemocyte functions. A** Volcano plot of deregulated genes upon depletion of dMi-2 in S2 cells. The -$\log_{10}$(p-value) is plotted against the $\log_2$ fold change of counts per gene in dMi-2-depleted (dsMi-2) vs. control cells (dsEGFP). Red dots represent significantly deregulated genes (adj. p < 0.01) obtained from biological triplicates (n = 3). **B** Gene ontology analysis of dMi-2-regulated genes. GO terms associated with hemocyte functions are highlighted in green. **C** Representative genes from all three gene classes regulated by Ush were analysed upon depletion of dMi-2 (dsMi-2) and Ush-B (dsUsh-B) by RT-qPCR. Gene names are indicated below. Expression was calculated relative to control treated cells (dsEGFP) and normalised using the mRNA levels of rp49. Error bars represent the standard deviation from biological replicates (n = 5) (T-test: *** p < 0.001, ** p < 0.01, * p < 0.05). Individual values of each replicate are displayed as circles.

to determine the contribution of Ush-B/dNuRD to these three transcription programmes. We depleted dMi-2 by RNAi and used a double stranded RNA specifically targeting the Ush-B isoform to deplete Ush-B (**Fig 3D**). We then measured changes to the transcriptome by RNA-seq. dMi-2 depletion led to significant changes in the levels of 945 transcripts (adj. p < 0.01; **Fig 6A**). A gene ontology analysis identified a number of GO terms associated with a wide range of biological processes (**Fig 6B**). These included "post-embryonic development" and "cell part morphogenesis" in agreement with the established role of dMi-2 in several differentiation processes [25–27]. GO terms related to the cell cycle or lipid metabolism were not strongly enriched. However, the GO term "innate immune response" was among the top 10 most strongly enriched GO terms.

Depletion of Ush-B had a comparatively mild impact on the transcriptome. 85 transcripts showed significant expression changes (adj. $p < 0.05$; **S5 and S6 Figs** and **S2 Table**). A significant fraction of these (18 genes, 21%) were either associated with GO terms related to immune response or macrophage function, have established roles in hemocyte biology or show specific expression in hemocytes. By contrast, only very few of the Ush-B regulated genes appeared to be involved in cell cycle regulation and/or could be related to metabolic pathways (**S5 and S6 Figs** and **S2 Table**).

We used RNAi and direct RT-qPCR to verify these results on representative hemocyte-related, metabolism and cell cycle genes (**Fig 6C**). Both depletion of Ush-B and dMi-2 resulted in increased expression of most hemocyte-related genes tested. By contrast, none of the genes encoding cell cycle regulators displayed drastic changes in expression after Ush-B or dMi-2 knockdown. Also, with the exception of fa2h which was upregulated upon dMi-2 depletion, and ACC which showed marginal expression changes in Ush-B or dMi-2 depleted cells, none of the lipid metabolism related genes responded to lowering the concentrations of Ush-B or dMi-2.

Taken together these results suggest that the Ush-B/dNuRD complex makes a contribution to the transcriptional programme that governs hemocyte functions but does not impinge on the transcriptional programmes regulating cell cycle and lipid metabolism. In a broader sense, these findings highlight how transcription cofactors make use of isoforms and isoform specific interactions with chromatin regulators to differentially regulate distinct gene expression programmes.

## Ush-B/dNuRD complex does not regulate the cell cycle

Unlike the simultaneous depletion of all Ush isoforms, the specific depletion of Ush-B or dMi-2 did not result in significant changes in the levels of cell cycle related transcripts. We, therefore, hypothesised that the Ush-B/dNuRD complex is not essential for cell proliferation. Indeed, neither isoform specific depletion of Ush-B, nor depletion of dMi-2 or the dNuRD subunit dMTA1-like produced the pronounced G2/M block observed following simultaneous depletion of all Ush isoforms (**Figs 3D** and **S7**). Although the percentage of cells in G2/M appeared to be somewhat increased and that of cells in G1 decreased these changes were not significant. Also, unlike simultaneous depletion of all Ush isoforms, depletion of Ush-B, dMi-2 or dMTA1-like did not alter protein expression levels of Cyclin A or Cyclin B (**Fig 3E**). We conclude that progression through the cell cycle does not rely on the Ush-B/dNuRD assembly. It is likely guided by other Ush isoforms or depends on redundant functions of several isoforms.

## Ush/dNuRD regulate hemocyte differentiation *in vivo*

While neither Ush-B nor dMi-2 depletion resulted in significant changes to genes encoding enzymes of the lipid metabolism and cell cycle genes, their depletion did lead to changes in the expression of several genes related to immune functions in the hemocyte-derived S2 cell line. This suggests that Ush-B/dNuRD contributes to the establishment and/or maintenance of specific functions of hemocytes. We, therefore, hypothesised that Ush-B/dNuRD might play a role in the regulation of hemocyte differentiation *in vivo*.

We have previously demonstrated that Ush restricts the activity of a *Hedgehog* enhancer in lymph glands, an important organ that limits hemocyte differentiation in L3 larvae [14,28,29]. Lymph glands are divided into a posterior signaling center (PSC), a medullary zone (MZ) containing hemocyte progenitors and a cortical zone (CZ) composed of differentiating and differentiated hemocytes. Cells in the PSC are secreting Hedgehog (Hh) which keeps the hemocyte

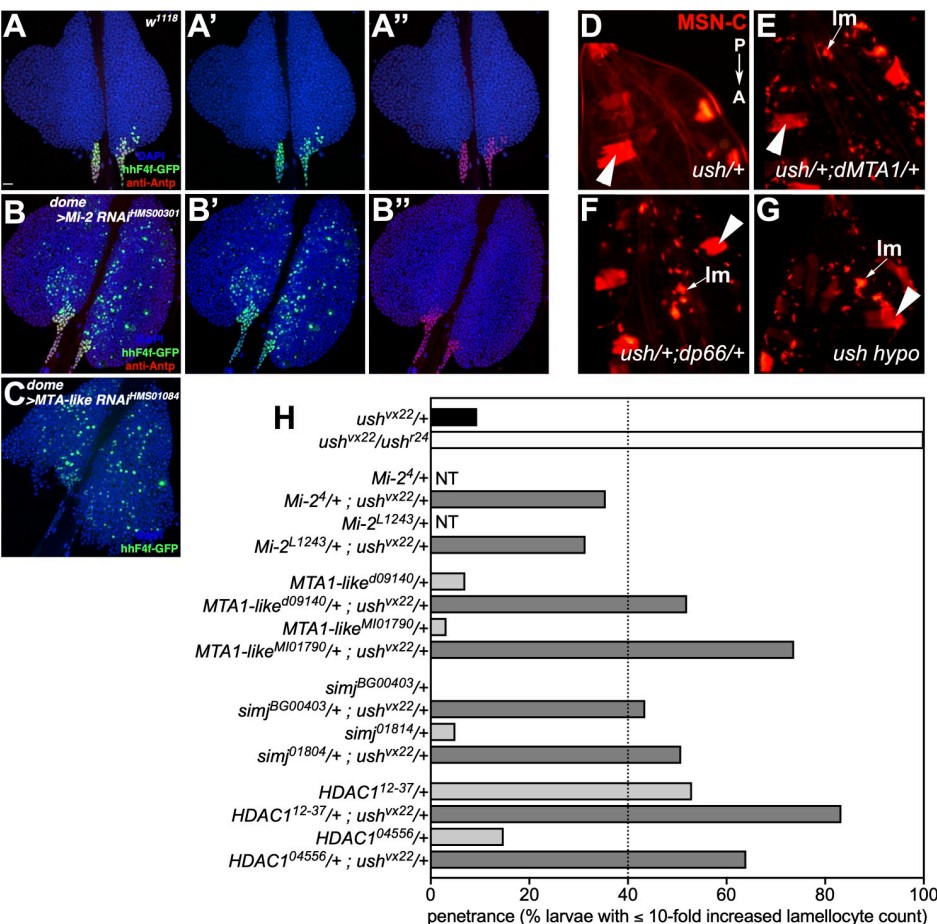

**Fig 7. Ush and NuRD regulate a hemocyte-specific enhancer and lamellocyte differentiation in *Drosophila* larvae.**
**A-C** Lymph glands isolated from wild type larvae (**A**), larvae that express a dsRNA against dMi-2 (**B**), or dMTA1-like
(**C**) under control of the domeless promoter (dome) active in the medullary zone. All larvae carry a construct,
reporting the activity of a minimal *Hedgehog* enhancer by GFP expression (hhF4f-GFP; green). PSC is marked using
immunostaining of Antennapedia (Antp; red). **D-H** Panels showing the dorsal view of the posterior region of late 3rd
instar larvae. The orientation is from top to bottom: posterior (P) to anterior (A). Lamellocyte (lm) differentiation is
blocked in ush heterozygotes (**D**). In contrast, ush/+;dMTA1-like/+ (**E**) and ush/+;dp66/+ (**F**) double heterozygotes
show lamellocyte differentiation. Likewise, ush^vx22/r24 (ush hypo; **G**) shows lamellocyte differentiation. Lamellocytes
express the MSN-cherry fluorescent transgene (MSN-C) and are marked with arrows. Larval muscles also express
MSN-C and are marked with large arrowheads. Penetrance of the lamellocyte differentiation phenotype was quantified
in **H**. Only larvae with a more than ten fold increased lamellocyte count were considered. Genotypes of respective
crosses are indicated below. For each NuRD allele two different mutant strains were tested as single heterozygotes
(light grey) or double heterozygous along with ush^vx22 (dark grey). Dashed line indicates the arbitrary cut-off for
identification of genetic interactors. NT: not tested.

progenitors in the MZ in a quiescent state and prevents their premature differentiation. *Hh*
expression in PSC cells is driven by an enhancer located in the first intron of the *Hh* gene [14].
Ush, which is expressed in the MZ but not in the PSC, is required for shutting off this enhancer
in the MZ and CZ, thereby limiting expression to the PSC. Accordingly, Ush loss of function
results in spurious *Hh* enhancer activity in the MZ and the CZ [14] (**S8 Fig**). We asked if
dNuRD, like Ush, was also involved in *Hh* enhancer repression. We used a fly strain carrying a
GFP reporter under control of an *Hh* enhancer fragment to address this question. In lymph
glands GFP activity is restricted to cells of the PSC as demonstrated by expression of the PSC
marker Antennapedia (Antp) (**Fig 7A**). We used the UAS/GAL4 system to deplete dMi-2 by

RNAi in the MZ (dome>dMi-2 RNAi). This resulted in the detection of GFP positive cells throughout the MZ and CZ (**Fig 7B**). Importantly, Antp expression remained restricted to the PSC demonstrating that PSC cells had not migrated into the MZ and CZ. We obtained the same result upon downregulation of the dNuRD subunit dMTA1-like (**Fig 7C**). These results establish that dNuRD is required to repress *Hh* enhancer activity in cells of the MZ and CZ and suggest that a Ush/dNuRD complex limits *Hh* expression to the PSC.

Next, we sought to determine if dNuRD cooperates with Ush to affect cell lineage decisions during hematopoiesis. In L3 larvae, Ush functions to suppress lamellocyte differentiation in absence of an appropriate trigger such as the injection of parasitic wasp eggs into the larva.

Whereas Ush hypomorphic (*ush^vx22/r24^*) larvae exhibit aberrant differentiation of progenitors into lamellocytes, a single wild type copy of Ush (*ush* heterozygotes) is sufficient to block lamellocyte differentiation [11,13]. In the past, we have exploited this situation to perform second-site non-complementation (SSNC) screens to identify factors that genetically cooperate with Ush in blocking lamellocyte differentiation [30,31]. In SSNC singular heterozygotes display a wild-type phenotype, whereas animals doubly heterozygous for two different genes exhibit a mutant phenotype.

We have constructed a fly stock that enables us to rapidly assay for SSNC with *ush* [28]. This stock carries a *ush* null allele (*ush^vx22^*) and the *misshapen-mCherry* (*MSN-C*) fluorescent reporter gene on the same chromosome. *MSN-C* is a marker for lamellocytes [32] and allowed us to rapidly identify larvae with increased numbers of lamellocytes using fluorescence microscopy. *MSN-C* is also constitutively active in larval muscle and serves as a marker for larvae that carry the *ush^vx22^*, *MSN-C* chromosome (**Fig 7D–7G**). In a screen setting, we routinely use an arbitrary level of at least 40% penetrance of the lamellocyte phenotype in double heterozygotes to identify robust genetic interactors. While this is less than penetrance levels typically observed for ush hypomorphs (70% to 100%), it is significantly greater than penetrance levels observed in negative controls (9.4%; **Fig 7H and S3 Table**). Here, we performed SSNC assays by combining *ush^vx22^* with mutant alleles of four dNuRD subunits: dMTA1-like, dp66 (simj), dRPD3 (HDAC1) and dMi-2 (**Fig 7H** and **S3 Table**). For each dNuRD subunit we carried out the assay with two independent mutant alleles. In all combinations we identified larvae with dramatically increased numbers of circulating *MSN-C*-positive lamellocytes (**Fig 7F and 7G**). Three of the four dNuRD complex subunits tested (dMTA1-like, dp66/simj and dRPD3/HDAC1), exhibited a greater than 40% penetrance when carried as double heterozygous with *ush* (**Fig 7H** and **S3 Table**). The two dMi-2 alleles exhibited 35% and 27% penetrance, respectively. While this was significantly greater than the control, it was less than the 40% penetrance we routinely use as a cut off.

We then tested if heterozygous alleles of the three dNuRD complex subunits that showed a robust genetic interaction with *ush* could produce lamellocytes when carried as singular heterozygotes in a *ush* wild-type background. Both alleles of dMTA1-like and dp66/simj exhibited minimal lamellocyte differentiation with a penetrance less than that of the control (**Fig 7H** and **S3 Table**). This strongly suggests that dMTA1-like and dp66 cooperate with Ush to block lamellocyte differentiation. In contrast, one of the two dRPD3/HDAC1 alleles tested exhibited a greater than 50% penetrance when carried as a singular heterozygote (**Fig 7H** and **S3 Table**). Currently, we do not understand the basis for this effect. Taken together with the fact that dRPD3 is not only a dNuRD subunit but exists in several other histone deacetylase complexes we cannot derive a clear conclusion as to the involvement of dRPD3 in lamellocyte differentiation. Nevertheless, the SSNC analysis identifies a robust genetic cooperation between *ush* and at least two dNuRD subunits, dMTA1-like and dp66, in blocking lamellocyte differentiation.

Taken together, our results demonstrate a function of Ush and dNuRD in regulating enhancer activity during hematopoiesis. Furthermore, we reveal that at Ush and dNuRD genetically cooperate in cell lineage commitment.

## Discussion

### Ush regulated transcription programmes

Ush genetically and physically interacts with GATA transcription factors to govern hemocyte differentiation during *Drosophila* hematopoiesis [6,7,13,14]. Ush has been demonstrated to modulate the expression of reporter genes and a small number of genes encoding hematopoietic regulators. However, the transcription programme controlled by Ush has not been defined on a genomewide level. We have determined genomewide binding sites of Ush and identified the genes regulated by Ush in the hemocyte-derived S2 cell line.

Ush binds more than 7,000 genomic locations and modulates the transcription of more than 1,800 genes. This demonstrates that, rather than being dedicated to the control of a small number of hematopoietic master regulators, Ush is a major regulator of the S2 transcriptome. We find that Ush bound regions are dramatically enriched for GATA sites on a genomewide level. This expands genetic and biochemical data that suggest that Ush cooperates with GATA transcription factors [6–8,13,14]. However, the binding sites for several other transcription factors are also strongly enriched in Ush bound regions. Interestingly, these include the E-box, a binding site for helix-loop-helix transcription factors. In mammals, composite GATA/E-box sites where the two elements are separated by 10 or less base pairs play a prominent role in determining lineage-specific gene expression during hematopoiesis [2]. These composite sites are bound by multisubunit transcription factor complexes containing GATA1, the Ush homolog FOG1, the basic helix-loop-helix transcription factors TAL1 and E47 and/or other hematopoietic regulators including LMO2 and LDB1. Our data suggest that similar composite GATA/E-box sites function in the *Drosophila* genome. In addition to GATA motifs and E-boxes, the GAGA and initiator sequences are present with high frequency in Ush-occupied regions. However, it remains to be demonstrated that Ush indeed forms complexes with these transcription factors and that together they regulate gene expression.

Our analysis of Ush regulated genes revealed that Ush modulates the expression of genes with hemocyte-related functions. S2 cells were derived from a primary culture of late embryos. They are believed to represent pro-hemocytes that are in the process of differentiating into macrophage-like plasmatocytes. Embryonic pro-hemocytes have the potential to either differentiate into plasmatocytes or into crystal cells [3]. Ush is expressed at high levels in pro-hemocytes but Ush expression is downregulated as pro-hemocytes differentiate into plasmatocytes and crystal cells. Differentiation into crystal cells relies on the expression of the Runx family transcription factor Lozenge (Lz). Our analysis reveals that reduction of Ush expression by RNAi in S2 cells derepresses *lz*. This suggests that the genetic suppression of crystal cell differentiation by Ush is based, at least in part, on its transcriptional repression of the crystal cell master regulator Lz [12].

Unexpectedly, we have also identified a large number of genes involved in lipid metabolism and cell cycle regulation that likewise require Ush to maintain their appropriate expression levels. Ush appears to be able to both positively and negatively affect gene regulation and it uses these opposing activities to coordinately regulate cellular functions at the transcriptional level.

### Ush regulates fatty acid metabolism

An illustrative example is provided by Ush's coordinated regulation of fatty acid metabolism. Several genes encoding enzymes of the beta-oxidation pathway that degrades fatty acids are

repressed by Ush. By contrast, the acetyl CoA carboxylase gene which encodes the key enzyme driving fatty acid synthesis requires Ush for its full expression. Accordingly, Ush appears to limit fatty acid degradation while it simultaneously promotes fatty acid synthesis. Interestingly, polarisation of mammalian macrophages is accompanied by the coordinated activation of fatty acid degradation in certain contexts [33]. This suggests that the transcriptional regulation of fatty acid metabolism by Ush might contribute to a metabolic profile that counteracts differentiation.

## Ush regulates the cell cycle

Ush also regulates cell cycle genes in a coordinated fashion. The RNA levels of several genes encoding proteins essential for the entry into and progression through mitosis are maintained at appropriate levels by Ush. This transcriptional regulation has functional significance since Ush depleted cells have strongly decreased proliferative capacity and exhibit a pronounced G2/M block. Ush is expressed in proliferating pro-hemocytes but downregulated in terminally differentiated plasmatocytes, crystal cells and lamellocytes [6]. We speculate that Ush supports the expansion of pro-hemocytes. Conversely, downregulation of Ush during lineage determination might allow these cells to exit the cell cycle for terminal differentiation. The human Ush homolog FOG1 has also been proposed to play a pro-proliferative role when overexpressed in NIH 3T3 cells [34,35]. However, this does not appear to involve the transcriptional regulation of cell cycle genes.

## Ush isoforms

Eukaryotes expand the diversity of their proteome by expressing multiple mRNA isoforms from the same protein coding gene. These can be generated by alternative splicing or the use of alternative transcriptional start sites. Indeed, more than 90% of human protein coding transcripts are estimated to be alternatively spliced. Functionally distinct isoforms of transcriptional regulators increase the capacity for fine-tuning transcriptional control. However, the molecular mechanisms by which different isoforms of transcriptional regulators contribute to gene expression are only beginning to be unravelled. Here, we have revealed that Ush is expressed in distinct isoforms that differ in their N-termini in S2 cells. We show that an N-terminal sequence unique to the Ush-B isoform mediates interaction with the dNuRD chromatin remodeling complex.

This dNuRD binding peptide is closely related to the FOG repression motif originally identified as a NuRD binding site in the mouse hematopoietic regulator FOG1 [15]. Related motifs are found in several other NuRD binding zinc finger proteins including FOG2, Sall4 and BCL11A. Importantly, dNuRD binds to both the N-terminus of Ush as well as to the N-terminus of FOG1. This demonstrates that this peptide based NuRD binding mechanism has been highly conserved in evolution.

Ush is the first *Drosophila* protein found to possess a dNuRD binding FOG repression motif. We have identified a second protein with an N-terminal FOG repression motif in the *Drosophila* proteome by sequence analysis, the O/E-associated zinc finger protein (OAZ). OAZ is not expressed in S2 cells and its relationship with dNuRD is unknown. Nevertheless, this finding hints that also in *Drosophila* the FOG repression motif is utilised in several proteins to mediate NuRD interaction.

In mammals FOG repression motif peptides have been shown to contact two NuRD subunits, RbAp46/RbAp48 and MTA1/2/3 but not the CHD4 ATPase [15,36]. Likewise, the *Drosophila* Mi-2 ATPase does not appear to directly bind the FOG repression motif given that the dMi-2-containing dMec complex does not bind to Ush or FOG repression motif peptides. We

propose that the FOG repression motif directly contacts dp55 (the homolog of RbAp46/RbAp48) and/or dMTA1-like. This hypothesis is supported by the observation that FOG repression motif mutations that disrupt binding to RbAp48 and MTA1/2/3 likewise abrogate binding to dNuRD [15,23].

It is interesting that the FOG1 peptide used in our study binds with higher affinity to dNuRD than the Ush peptide even though the FOG repression motif contained within both peptides is identical. This suggests that amino acids outside of the FOG repression motif contribute to dNuRD binding. It is also possible that interaction between dNuRD and Ush is modulated by post-translational modification within or in the vicinity of the FOG repression motif *in vivo*. Indeed, phosphorylation of serine residue 2 within the FOG repression motif has previously been shown to lower NuRD binding [37].

## Impact of Ush-B/dNuRD on transcription in S2 cells

Specific depletion of Ush-B by RNAi had a mild effect on the S2 transcriptome compared to the simultaneous depletion of all Ush isoforms. In principle, it is possible that Ush-B occupies a smaller set of genomic loci compared to other isoforms. We consider this to be unlikely. A large number of Ush bound regions contains binding sites for GATA transcription factors that have been implicated in recruiting Ush to chromatin. GATA transcription factors interact with zinc fingers that are shared in all Ush isoforms which should, therefore, be recruited equally well to all GATA transcription factor occupied sites. We consider it more likely that Ush-B and other Ush isoforms both contribute to gene regulation. In this scenario, Ush-B depletion does only change the transcript levels of genes that require high concentrations of Ush for their repression or that are particularly dependent on Ush-B and the Ush-B/dNuRD complex. Many of these Ush-B depletion-sensitive genes have hemocyte-related functions. Indeed, progressive downregulation of Ush drives gene expression changes that are required for the differentiation of specialised hemocytes such as plasmatocytes, crystal cells and lamellocytes *in vivo*. Unlike cell cycle and metabolism genes, these genes appear to be uniquely sensitive to modest reduction of overall Ush expression levels obtained by selective depletion of Ush-B. Moreover, these genes are also repressed by dMi-2 suggesting that they are, indeed, targets of the Ush-B/dNuRD complex. This suggests that the Ush-B/dNuRD complex is particularly important for regulating the transcription of genes characteristic for macrophage function.

## Ush and dNuRD cooperate in hemocyte differentiation *in vivo*

Hematopoiesis in *Drosophila* occurs at various developmental stages including embryogenesis and larval development. Our results have revealed that Ush and dNuRD mould the metabolism, proliferation and hemocyte-related functions of S2 cells by maintaining an extensive gene expression programme. S2 cells are derived from embryonic hemocytes indicating gene regulatory roles for Ush and dNuRD during embryonic hematopoiesis. Our genetic loss-of-function analyses show that Ush and dNuRD also regulate hematopoiesis at later developmental stages. In particular, we have shown that Ush and dNuRD subunits are required to restrict *Hedgehog* enhancer activity to cells of the posterior signaling center in lymph glands of L3 larvae. This result suggests that Ush and dNuRD actively modulate gene expression programmes also at the larval stage. Moreover, Ush and dNuRD suppress lamellocyte differentiation in unstressed larvae. We do not yet know to which extent the different Ush isoforms are required for lamellocyte suppression. However, the finding that mutations in dNuRD subunits result in excessive lamellocyte differentiation only in a genetic background with reduced Ush activity demonstrates genetic cooperativity between Ush and dNuRD. This is consistent with the hypothesis that a Ush-B/dNuRD complex is active during larval hematopoiesis.

The function of dNuRD in hematopoiesis identified by our work solidifies the important role of this complex as a regulator of differentiation in *Drosophila*. We have previously shown that dMi-2 cooperates with transcription factors such as Tramtrack 69 or Kumgang to determine cell lineages in different developmental settings ranging from neurogenesis to spermatogenesis [25–27,38]. In each of these scenarios a different lineage-specific transcriptional regulator (Tramtrack 69, Kumgang, Ush) utilises the ubiquitously expressed dMi-2 complex to establish lineage- and stage-appropriate gene expression programmes.

Although the process of hematopoiesis in *Drosophila* is far less complex than in mammals, Ush and FOG1 play remarkably similar roles in suppressing certain hematopoietic lineages. FOG1 facilitates erythroid and megakaryocyte differentiation while suppressing mast cell differentiation. While high Ush levels in hemocyte progenitors counteracts differentiation into all three *Drosophila* hemocyte cell types, intermediate Ush levels are sufficient to suppress crystal cell and lamellocyte differentiation but compatible with differentiation of plasmatocytes [10]. Both FOG1 and Ush cooperate with NuRD using a highly conserved short peptide motif. Thus, our study identifies the FOG1/Ush-NuRD complex as an ancient component of the machinery regulating hematopoiesis.

Cell lineage differentiation relies on a finely orchestrated series of events that change cell morphology and function at multiple levels. These include division of stem cells, the proliferation of progenitors, their withdrawal from the cell cycle for terminal differentiation, the timely expression of lineage-specific genes and the generation of changing metabolic profiles that are appropriate for each stage of differentiation. By coordinately regulating the transcription of cell cycle genes, genes encoding metabolic enzymes and genes performing macrophage-specific functions Ush simultaneously controls several cellular activities that are relevant to the differentiation process. A classical 'master regulator' of differentiation sits on top of a hierarchy and directs the expression of downstream transcription factors that in turn generate gene expression profiles committing cells to a certain lineage. By contrast, Ush appears to be more "hands-on" and directly regulates the expression of different types of genes that are key for diverse processes impinging on differentiation.

## Materials and methods

### Cell culture

*Drosophila melanogaster* S2 and S2[Cas9] cells (S2 cells expressing the Cas9 nuclease from Streptococcus pyogenes; generous gift from Klaus Förstemann, Munich) were cultured in Schneider's *Drosophila* Medium (2172001, Gibco) supplemented with 10% (v/v) fetal bovine serum (FBS; F7524, Sigma) and 1% (v/v) Penicillin-Streptomycin (15140122, Gibco). Cell lines were grown under standard conditions at 26˚C.

### Endogenous tagging using CRISPR/Cas9

CRISPR/Cas9-based insertion of epitope-tag sequences into the genome of *Drosophila* S2 cells was performed as previously described (Bottcher et al., 2014). DNA sequences coding for GFP- or FLAG-tags were inserted at the 3' end of the coding region of the U-shaped or dMi-2 gene locus, leading to expression of C-terminally tagged proteins.

In brief, S2 cells stably expressing the Cas9 nuclease (S2[Cas9] cells) were transfected with double stranded linear DNA constructs (1) encoding for sgRNA and (2) providing a template for homologous recombination (HR). Both of these constructs were generated by PCR using gene specific primers (S4 Table). The sgRNA sequences were designed to target Cas9 as close to the respective STOP codon as possible with respect to the nearest available protospacer adjacent motif (PAM) (targeting sequences: CATTTGAGAAAGCCAGCTG (Ush) and

TCGAATAATTCCGGCGTCT (dMi-2)). Homologous recombination templates were amplified from plasmids containing GFP- or FLAG-tag sequences including a STOP codon as well as a resistance marker under control of a copia promoter. This insert was amplified using primers containing 60 bp sequences homologous to regions directly up- and downstream of the original STOP codon. In particular, HR templates for C-terminal tagging of U-shaped were amplified using the following plasmids: pSK23 (GFP-tag & Puromycin resistance marker; Addgene #72851) and pSK25 (2xFLAG-tag & Puromycin resistance marker; Addgene #72853). HR templates for C-terminal tagging of dMi-2 were amplified using the following plasmids: pMH3 (GFP-tag & Blasticidin resistance marker; Addgene #52528) and pMH4 (2xFLAG-tag & Blasticidin resistance marker; Addgene #52529).

To favour double strand break repair by HR, the protein amount of key enzymes involved in non-homologues end joining (NHEJ) and microhomology-mediated end joining (MMEJ) was lowered by transfecting S2[Cas9] cells with 1 µg/ml dsRNA targeting lig4 (NHEJ) and mus308 (MMEJ) transcripts. After three days, cells were transfected with HR and sgRNA templates using FuGENE HD transfection reagent (E2311, Promega). Four days post transfection cells were transferred to medium containing 2 µg/ml Puromycin (540411, Merck) or 10 µg/ml Blasticidin (A11139, Gibco) respectively. Cells were kept under selection for at least 14 days or until non-resistant control cells declined.

To retrieve monoclones, cells were serially diluted in 96 well plates. Monoclones were expanded and screened by PCR on genomic DNA using primers flanking the insertion site.

## RNA interference in *Drosophila* S2 cells, proliferation and viability assay

Double-stranded RNA (dsRNA) was synthesised using the MEGAscript T7 kit (AMB1334, Invitrogen) according to manufacturer's instructions. In brief, dsRNA was generated using T7 Polymerase *in vitro* transcription from PCR amplicons obtained with T7 minimal promotor containing primers using a cDNA template from S2[Cas9] cells. 10-15 µg of dsRNA was added to $0.3x10^6$ S2[Cas9] cells in a total of 3 ml Schneider's *Drosophila* Medium. For different cell numbers, the amount of dsRNA and medium was scaled accordingly. Cells were harvested for RNA isolation four days post transfection and for cell cycle analysis and protein extraction three days post transfection.

To monitor proliferation, cells were re-seeded immediately after transfection. The cell density was determined from three independent dsRNA transfections every 24 hours using a hemocytometer. Cell viability was determined four days post transfection by measuring cell dilutions on a CASY Cell Analyser (OMNI Life Science).

## Cell cycle analysis by flow cytometry

Cell cycle distribution of *Drosophila* cell lines was analysed as described in [39] with minor changes. In brief, cells were harvested, washed and resuspended in 500 µl PBS. While vortexing cells were fixed by the addition of 5 ml ice cold 95% (v/v) ethanol. One day prior to analysis cells were rehydrated in PBS for 5 min on ice, washed and finally resuspended in 1 ml PBS. 25 µl of RNAse A digestion mix (10 mM PIPES/NaOH pH 6.8, 100 mM NaCl, 2 mM $MgCl_2$, 0.25 mM EDTA, 0.2% (w/v) Triton X-100, 100 µg/µl RNAse A) and 50 µl propidium iodide solution (0.5 mg/ml propidium iodide in 38 mM sodium citrate) were added and DNA was stained overnight at 4˚C with rotation.

Flow cytometry was performed on an ARIA III cytometer (BD) with DIVA 8.0.2 software. After gating the cells of interest in an FSC-A/SSC-A plot debris and doublets were excluded with an PE-Area vs. PE-Width Plot. Measurements were taken from three independent dsRNA transfections where 10,000 cells were counted per replicate. For visualisation and

record of the PI-signal a histogram for the PE-channel (excitation 561 nm) was used with a 582/15 bandpass filter. For analysis of the recorded signals the exported fcs (3.0) files were loaded in FlowJo (10.6.1). The Watson Pragmatic algorithm was used for computation of G1, S and G2/M fractions [40].

## Preparation of protein extracts

For whole cell extracts cells were washed in PBS and lysed in RIPA buffer (50 mM Tris/HCl pH 8.0, 150 mM NaCl, 1 mM EDTA, 1 mM EGTA, 1% (w/v) NP-40, 0,5% (w/v) sodium deoxycholate, 0,1% (w/v) SDS, 10% (v/v) glycerol, 1 mM DTT) for 20 min with rotation at 4˚C followed by freeze/thaw lysis in liquid nitrogen. Lysates were cleared by centrifugation at 21,100 g and 4˚C for 20 min. The protein content was determined using DC Protein Assay (5000112, Biorad) according to manufacturer's instructions.

Nuclear extracts were obtained by washing cells in PBS followed by hypotonic lysis in buffer B (10 mM Hepes/KOH pH 7.6, 10 mM KCl, 1.5 mM $MgCl_2$, 1 mM DTT) for 15-20 min with rotation at 4˚C. Nuclei were pelleted by centrifugation at 4,500 g and 4˚C for 15 min. Nuclear proteins were extracted in buffer C (20 mM Hepes/KOH pH 7.6, 420 mM NaCl, 1.5 mM $MgCl_2$, 0.2 mM EDTA, 25% (v/v) glycerol, 1 mM DTT) for 30 min with rotation at 4˚C. Extracts were cleared by centrifugation at 21,100 g and 4˚C for 45 min. The protein content was determined via Bradford method using Protein Assay (5000006, Biorad) according to manufacturer's instructions.

Nuclear extract from *Drosophila* embryos (TRAX) was obtained as previously described [41].

## RT-qPCR and RNA-seq

Total RNA was isolated using the peqGOLD Total RNA Kit (12-6834-02, Peqlab) together with the peqGOLD DNase I Digestion Kit (732-2982, Peqlab) and the integrity of RNA was evaluated on a 1.2% Agarose/TAE gel. For RT-qPCR cDNA was prepared from 1 μg of total RNA using the SensiFAST cDNA Synthesis Kit (BIO-65054, Bioline) and analysed by qPCR using the SensiFast SYBR Lo-ROX Kit (BIO-94050, Bioline) according to manufacturer's instructions together with gene-specific primers (S4 Table). Amplification reactions were measured in triplicates on a Stratagene Mx3000P thermocycler (Agilent Technologies) and the mean values were calculated according to the ΔΔCt method using the mRNA levels of Rp49 as a normalisation reference. mRNA expression was calculated relative to samples treated with a non-targeting dsRNA against GFP. Error bars represent the standard deviation from five biological replicates.

For RNA sequencing the total RNA from three independent dsRNA transfections was isolated. The integrity of RNA was assessed on an Experion StdSens RNA Chip (Bio-Rad). RNA-seq libraries were prepared using a TruSeq Stranded mRNA Library Prep kit (Illumina). Libraries were quantified on a Bioanalyzer (Agilent Technologies) and sequenced on an Illumina HiSeq 1500 platform, rapid-run mode, single-read 50 bp (HiSeq SR Rapid Cluster Kit v2, HiSeq Rapid SBS Kit v2, 50 cycles) according to the manufacturer's instructions.

## SDS-PAGE and Western blot

Proteins were electrophoretically separated on a SDS-polyacrylamid gel (SDS-PAGE) and then transferred onto activated polyvinylidene difluoride (PVDF) membranes (T830.1, Roth) by Western Blotting in Pierce Western Blot Transfer Buffer (35040, Thermo Fisher Scientific). Membranes were saturated in Blocking buffer (PBS, 0.1% (w/v) Tween-20, 5% (w/v) non-fat dry milk) for 1 h at room temperature and subsequently incubated with the respective

antibody dilution in Blocking buffer overnight at 4˚C. After washing the membranes four times for 5 min at room temperature in Washing buffer (PBS, 0.1% (w/v) Tween-20) appropriate HRP-coupled secondary antibodies (anti-mouse IgG (NA931, GE Healthcare), anti-rabbit IgG (NA934, GE Healthcare), anti-rat IgG (31470, Thermo Fisher Scientific), anti-guinea pig IgG (706-035-148, Jackson ImmunoResearch)) were applied in Blocking buffer for 2 h at room temperature. After four washing cycles for 5 min in Washing buffer Western blot signals were detected by chemiluminescence using the Immobilon Western Blot Chemiluminescence HRP substrate (WBKLS0500, Millipore).

Antibodies and antisera were used in the following dilutions: Ush (1:5,000; (Fossett et al., 2001)), GFP (1:5,000; clone [3H9] from Chromotek), FLAG (1:8,000; clone M2 from Sigma), Tubulin beta (1:8,000; clone KMX-1 from Merck Millipore), dMi-2 (1:8,000; [42]), dMTA1--like (1:10,000; [38]), Cyclin B (1:5,000; clone F2F4 from DHSB), Cyclin A (1:1,000; clone A12 from DHSB), Lamin Dm0 (1:5,000; clone ADL67.10 from DHSB), dp66 (1:10,000; [43]), dp55 (1:20,000; [44]), dMEP-1 (1:10,000; [38]), dRPD3 (1:10,000; [42]), dCHD3 (1:10,000; [45]), dPc (1:50,000; [46]), dE(z) (1:1,000; [47]), dLSD1 (1:5,000; [48]).

## Peptide Synthesis and usage in competition assays

Peptides were synthesised in a 10 µmol scale (0.25 mmol/g) following the standard solid phase peptide synthesis (SPPS) methodology, using Fmoc-amino acids and Oxyma/DIC as coupling agents. Final deprotection and cleavage from the solid support was performed with 1.5 ml of cleavage cocktail: 94 TFA/1 TIS/ 2.5 DODT/2.5 $H_2O$ for 3 h. Obtained peptides were purified at 25˚C by preparative reverse phase (RP)-HPLC performed on a PLC 2020 personal purification system (Gilson) with a preparative Nucleodur C18 HTec-column (5 µm, 250 × 16 mm; Macherey Nagel) and a flow rate of 10 ml/min. Detection of the signals was achieved with a UV detector at 220 nm wavelength. The eluents were MilliQ $H_2O$ and MeCN with addition of 0.1% TFA applied at a gradient of 5-40% MeCN.

Peptides were diluted and concentrations were determined according to [49]. The following concentrations were used in interaction assays: 3.5 µM, 7.0 µM, 14.0 µM (FOG1 peptides) and 17.5 µM, 35 µM (Ush peptides) in GST pulldown assays; 1.0 µM, 2.0 µM, 3.0 µM FOG1 peptides in immunoprecipitation assays.

## Co-Immunoprecipitation of epitope-tagged proteins

1 mg of nuclear extract was diluted 1:4.2 with buffer C-0 (20 mM Hepes/KOH pH 7.6, 1.5 mM $MgCl_2$, 0.2 mM EDTA, 25% (v/v) glycerol, 0.131% (w/v) NP-40, 1 mM DTT) and adjusted to 1 ml final volume with buffer C-100 (20 mM Hepes/KOH pH 7.6, 100 mM NaCl, 1.5 mM $MgCl_2$, 0.2 mM EDTA, 25% (v/v) glycerol, 0.1% (w/v) NP-40, 1 mM DTT). 5 U/ml of Benzonase was added (70664, Millipore), samples were incubated for 1 h at 4˚C with rotation and diluted extracts were cleared of contingent precipitates by centrifugation (15 min, 21,100 g, 4˚C). 25 µl of GFP-Trap Agarose (gta, ChromoTek) or ANTI-FLAG M2 Affinity Gel (A2220, Sigma) was blocked in buffer C-100 containing 1 mg/ml BSA and 1% (w/v) fish skin gelatin for 1 h at 4˚C with rotation and then added to the diluted extracts. Immunoprecipitation was carried out overnight at 4˚C with rotation. The resin was washed four times with 1 ml IP150 buffer (25 mM Hepes/KOH pH 7.6, 150 mM NaCl, 12.5 mM $MgCl_2$, 0.1 mM EDTA, 10% (v/v) glycerol 0.1% (w/v) NP-40, 1 mM DTT) and finally resuspended in SDS-PAGE loading buffer (50 mM Tris/HCl pH 6.8, 2% (w/v) SDS, 10% (v/v) glycerol, 0.1% (w/v) bromophenol blue, 100 mM DTT). Immunoprecipitates were analysed by SDS-PAGE and Western blot.

## Chromatin Immunoprecipitation followed by next-generation sequencing (ChIP-seq)

$10^8$ S2[Cas9] cells expressing endogenously tagged proteins were cross-linked with 1% Formaldehyde for 10 min at RT with agitation. Fixation was quenched by addition of Glycin to a final concentration of 240 mM and incubation for 10 min at RT with agitation. After two times washing in PBS cells were lysed in 1 ml of ChIP Lysis buffer (50 mM Tris/HCl pH 8.0, 10 mM EDTA, 1% (w/v) SDS, 1 mM DTT) for 10 min on ice. Chromatin was sheared by sonication in the Bioruptor UCD-200TM-EX (Diagenode) supplied with ice water in three cycles over 30 min. Each cycle lasted for 10 min with 10x 30 s intervals of sonication at high power followed by 30 s without sonication to ensure proper cooling. Cell debris were pelleted by centrifugation (20 min, 21,100 g, 4˚C) and the supernatant containing fragmented chromatin was stored at -80˚C. The fragment size was monitored by decrosslinking 50 µl of chromatin-containing lysate in presence of RNase A (400 ng/µl; A3832, Applichem) and Proteinase K (400 ng/µl; 7528.1, Roth) for 3 h at 55˚C followed by 65˚C overnight. DNA was purified using the QIAquick PCR Purification Kit (28106, Qiagen) and fragment sizes were evaluated on a 1.2% Agarose/TAE gel.

For one ChIP reaction 140 µl of chromatin lysate was pre-cleared by diluting it 1:10 in ChIP IP buffer (16.7 mM Tris/HCl pH 8.0, 1.2 mM EDTA, 167 mM NaCl, 1.1% (w/v) Triton X-100, 0.01% (w/v) SDS, 1 mM DTT) and addition of 40 µl Protein A Sepharose resin (nProtein A Sepharose 4 Fast Flow, 17-5280, GE Healthcare) that had been blocked for 1 h in ChIP Blocking buffer (ChIP Low salt buffer containing 2 mg/ml BSA and 2% (w/v) fish skin gelatin). After incubation at 4˚C for 1 h with rotation, beads were collected (centrifugation for 10 min at 21,100 g and 4˚C) and the supernatant was added to 25 µl of blocked GFP-Trap Agarose (gta, ChromoTek).

Immunoprecipitation (IP) took place overnight at 4˚C with rotation followed by extensive washing: Three times with 1 ml of ChIP Low salt buffer (20 mM Tris/HCl pH 8.0, 2 mM EDTA, 150 mM NaCl, 1% (w/v) Triton X-100, 0.1% (w/v) SDS, 1 mM DTT), three times with 1 ml of ChIP High salt buffer (20 mM Tris/HCl pH 8.0, 2 mM EDTA, 500 mM NaCl, 1% (w/v) Triton X-100, 0.1% (w/v) SDS, 1 mM DTT), once with 1 ml of ChIP LiCl buffer (10 mM Tris/HCl pH 8.0, 1 mM EDTA, 250 mM LiCl, 0.1% (w/v) NP-40, 1 mM DTT) and finally twice with TE buffer (10 mM Tris/HCl pH 8.0, 1 mM EDTA). Each washing step was carried out at 4˚C for 5 min with rotation and the resin was pelleted in between by centrifugation (4 min, 400 g, 4˚C).

Cross-linked protein-DNA complexes were eluted twice from the resin in 250 µl ChIP elution buffer (100 mM $NaHCO_3$, 1% (w/v) SDS) for 20 min at RT with rotation. The resin was pelleted by centrifugation (3 min, 1,200 g, RT) and the eluate was removed. After the second elution cycle the resin-buffer suspension was incubated at 95˚C for 10 min, the resin was pelleted and both eluates were pooled. 14 µl of pre-cleared chromatin was added to 500 µl of ChIP elution buffer as "input" sample. 40 µM of NaCl was added to IP and input samples and protein-DNA complexes were decrosslinked overnight at 65˚C with agitation. 40 mM Tris/HCl pH 6.8, 1 mM EDTA and 40 ng/µl Proteinase K (7528.1, Roth) was added to each sample and proteins were digested at 45˚C for one hour with agitation. The DNA was purified using QIAquick PCR purification kit (28106, Qiagen).

Purified DNA from up to six ChIP reactions was pooled, concentrated (Concentrator 5301, Eppendorf) and quantified using the Qubit dsDNA High-Sensitivity Assay Kit (Q32851, ThermoFisher scientific). Libraries were generated from 1 ng of DNA using the MicroPlex Library Preparation Kit v2 (C05010012, Diagenode) according to manufacturer's instructions. The amplified libraries were purified using AMPure XP beads (A63880, Beckman Coulter) and eluted in TE buffer.

The quality of sequencing libraries was controlled on a Bioanalyzer 2100 using the Agilent High Sensitivity DNA Kit (Agilent). Pooled sequencing libraries were quantified with digital polymerase chain reaction (PCR) (QuantStudio 3D, Thermo Fisher) and sequenced on an Illumina HiSeq 1500 platform, rapid-run mode, single-read 50 bp (HiSeq SR Rapid Cluster Kit v2, HiSeq Rapid SBS Kit v2, 50 cycles) according to the manufacturer's instructions.

## GST pulldown assay

pGEX2T-mFOG1(1-45) [15] or pGEX4T1 expression constructs were transformed into an E. coli BL21DE3 strain (C2527H, NEB). The culture was expanded and expression was induced at an $OD_{600}$ of 0.7 with 0.4 mM IPTG. After 24 h at 18°C bacteria were harvested, washed with PBS and resuspended in PBS/Triton (PBS containing 1% (w/v) Triton X-100). For lysis, cells were sonicated 12 times for 12 s on an ultrasonic homogenizer (HD2200, Bendelin electronics) at 25% output while keeping the suspension on ice in between. The suspension was frozen in liquid nitrogen and thawed on ice three times before cell debris were pelleted by centrifugation at 4°C and 27,000 g for 30 min. GST-fusion proteins were coupled to Glutathione Sepharose 4 Fast Flow (17-5132-01, GE Healthcare) for 2 h at 4°C with rotation. Unbound proteins were removed by washing three times with PBS/Triton and twice with PBS for 5 min at 4°C with rotation. The amount of GST-fusion protein bound to the Sepharose resin was evaluated by comparison to a BSA standard on a Coomassie stained SDS-PA gel.

GST pulldown interaction assays were performed using 10-20 µg of GST fusion proteins and 1 mg of S2 cell nuclear extract or TRAX per pulldown reaction. The resin was blocked for 1 h at 4°C with rotation in GST Pulldown Buffer containing 1 mg/ml BSA and 1% (w/v) fish skin gelatin. Binding took place overnight at 4°C with rotation in 1 ml GST Pulldown buffer (25 mM Hepes/KOH pH 7.6, 150 mM KCl, 12.5 mM $MgCl_2$, 0.1 mM EDTA, 20% (v/v) glycerol 0.1% (w/v) NP-40, 1 mM DTT). The resin was washed four times with 1 ml GST Pulldown buffer for 5 min at 4°C with rotation followed by centrifugation (4 min, 1,500 g, 4°C). Interacting proteins were analyzed by SDS-PAGE and Western blot.

## Fly stocks

The $w^{1118}$ served as the wild-type control. The following stocks were obtained from the Bloomington stock center: $w^{1118}$;MTA1-like$^{d09140}$/TM6B,Tb$^1$, y$^1$ w*;MTA1-like$^{MI01790}$, w$^{1118}$; simj$^{BG00403}$/TM6B, Tb$^1$, simj$^{01814}$ ry$^{506}$, y$^1$ w*;HDAC1$^{12-37}$/TM6B,Tb$^1$, HDAC1$^{04556}$ ry$^{506}$/TM3,ry$^{RK}$ Sb$^1$ Ser$^1$, Mi-2$^4$ red$^1$ e$^4$/TM6B, Sb$^1$ Tb$^1$ ca$^1$, y$^1$ w$^{1118}$; Mi-2$^{L1243}$/TM3, Ser$^1$. The dome-GAL4 line was a gift from U. Banerjee (UCLA). y w$^{67c23}$; ush$^{vx22}$/CyO y$^+$ and y w$^{67c23}$; ush$^{R24}$/CyO y$^+$, the misshapen-mCherry (MSN-C) and the hhF4f-GFP fluorescent reporter transgene stocks have been described previously [11,14,32]. The y w;ush$^{VX22}$, MSN-C/CyO y$^+$ was created using standard recombination procedures. Larvae were cultured at 23°C and late 3rd instar wandering larvae were assayed for lamellocyte differentiation. Fluorescent microscopy was conducted using a Zeiss Axioplan microscope.

## Second site non complementation assays

Larvae were cultured at 23°C and late 3$^{rd}$ instar wandering larvae were assayed for lamellocyte differentiation. Larvae were placed on a slide with a drop of PBS and observed under fluorescent microscopy using a Zeiss Axioplan microscope. Only larvae with *MSN-C* fluorescent reporter transgene expression were scored.

### *Hh* enhancer reporter assay in larval lymph glands

The dome-Gal4 line was crossed with appropriate hhF4f-GFP;UAS-RNAi lines, mid-third instar larvae were collected and lymph glands were dissected lymph glands. Immunostaining was performed as described previously [14]. The following antibodies were used to identify PSC cells: mouse anti-Antp (primary antibody; 1:100; 4C3, Developmental Studies Hybridoma Bank); Alexa 555-conjugated mouse IgG antibody (secondary antibody; A28180, Thermo Fisher Scientific). Cell nuclei were stained with DAPI (Invitrogen). Immunostained samples were analysed with a Nikon A1R laser-scanning confocal microscope.

### Bioinformatical analysis

ChIP-Seq data were aligned to *Drosophila* Genome dm3, using bowtie2 [50]. Bigwig files were obtained using Galaxy/deepTools [51] normalised to genome Coverage. Data were visualised in the UCSC genome browser [52]. Data analysis was performed using Galaxy [53], Cistrome [54] and Bioconductor/R [55]. Peaks were identified using MACS2 [56] with the following settings: Set lower mfold bound = 5; Set upper mfold bound = 50; Band width for picking regions to compute fragment size = 300; Peak detection based on = q-value; Minimum FDR = 0.05. Overlap between peaks was obtained using the Venn Diagram tool within Galaxy/Cistrome platform. Peaks were considered overlapping at $\geq 1$ common nucleotide. Enriched motifs were identified using HOMER [57]. Heatmaps were obtained using Galaxy/deepTools. Overlap with genomic features was determined using "CEAS: Enrichment on chromosome and annotation" [58] within the Galaxy/Cistrome platform. Profiles of the histone marks were obtained using Galaxy/deepTools. Following public datasets were used: H3K4me1 (GSM2259983, GSM2259984), H3K4me3 (GSM2259985, GSM2259986), H3K27ac (GSM2259987, GSM2259988) [59], H3K27me3 (GSM2776903) [60], Mi-2 modeENCODE (GSM1147259, GSM1147260), Mi-2 (ERR1331728, ERR1331729) [26]. Transcription start site (TSS) annotation was obtained from the UCSC table browser and coverage profiles were calculated using Galaxy/deepTools.

RNA-Seq data were aligned to *Drosophila* transcriptome using RNA Star (2.7.2b) [61]. Counts per gene were determined using FeatureCounts (1.6.4) [62]. Differentially expressed genes and normalised reads were determined using DeSeq2 (2.11.40.6) [63]. Gene ontology analysis on significantly deregulated genes (adj. p < 0.01) was performed using the Metascape tool (version 3.5, 2019-08-14, [64]) on "Express Analysis" settings. Additional GO terms and transcript expression patterns were obtained from FlyBase (version FB2019_06) and the Berkley *Drosophila* Genome Project (release 3, 2019-06-04) respectively.

## Supporting information

**S1 Fig. Insertion of GFP- or FLAG-tag sequences at *Ush* and *dMi-2* 3' ends using CRISPR/ Cas9. A** Schematic representation of the *Ush* gene locus before (top) and after insertion of GFP (middle) and FLAG (bottom) tagging constructs. Black boxes represent exons, black (broken) lines represent introns. The inserted tag sequences (GFP: green, FLAG, red) and selection marker (promoter: ochre, Puromycin resistance: orange) are highlighted. The positions of primers used for genotyping of Ush alleles are indicated with purple arrowheads. **B** PCR from genomic DNA of control cells and cells modified to express GFP- or FLAG-tagged Ush, respectively. Insertion of the tag sequence followed by a Puromycin selection marker is monitored using primers surrounding the 3' end of the coding region within the Ush gene. Non-tagged alleles give rise to a 216 bp amplicon, GFP- and FLAG-tagged alleles result in 1991 bp and 1311 bp fragments respectively. **C** Schematic representation of the *dMi-2* gene locus before (top) and after insertion of GFP (middle) and FLAG (bottom) tagging constructs. Black boxes

represent exons, black (broken) lines represent introns. The inserted tag sequences (GFP: green, FLAG, red) and selection marker (promoter: light blue, Blasticidin resistance: dark blue) are highlighted. The positions of primers used for genotyping of Ush alleles are indicated with purple arrowheads. **D** PCR from genomic DNA of control cells and cells modified to express GFP- or FLAG-tagged dMi-2, respectively. Insertion of the tag sequence followed by a Blasticidin selection marker is monitored using primers surrounding the 3' end of the coding region within the Ush gene. Non-tagged alleles give rise to a 200 bp amplicon, GFP- and FLAG-tagged alleles result in 1737 bp and 1077 bp fragments respectively. **E** Nuclear extracts of control cells and cells expressing endogenously tagged dMi-2-GFP or dMi-2-FLAG was probed on Western blot using antibodies against dMi-2, GFP or FLAG. Tubulin signal serves as loading control.
(TIF)

**S2 Fig. Ush occupancy at the *lozenge* and the *atilla* gene locus. A** Genome browser snapshots of the *lozenge* (*lz*) (top) and the *atilla* (bottom) gene locus displaying Ush occupancy (green) determined by Ush-GFP ChIP-seq. Input signals are shown in black. Location of genes is displayed below with boxes indicating exons.
(TIF)

**S3 Fig. Expression of Ush isoforms in S2 cells. A** Genome browser snapshots of the Ush gene locus displaying RNA-seq coverage in S2 cells from biological triplicates. Exons encoding unique N-termini are highlighted in green (Ush-B specific) and orange (Ush-A specific).
(TIF)

**S4 Fig. Comparison of dMi-2 ChIP-seq datasets. A** dMi-2 ChIP-seq peaks obtained in this study were ranked and signals were compared to two other datasets (Kreher et al., 2017 and modENCODE ID 5070) in a region of 5 kb surrounding the respective peak. **B** Genome browser snapshots of an exemplary region displaying dMi-2 occupancy (red: this study; ochre: Kreher et al., 2017; blue: modENCODE ID 5070). Input signals of this study are shown in black. Location of genes is displayed below with boxes indicating exons.
(TIF)

**S5 Fig. Ush-B repressed genes.** Tables of genes that are significantly upregulated (adj. p < 0.05) upon depletion of of Ush-B. Gene symbols are indicated along with the respective fold change relative to cells transfected with control dsRNA (dsEGFP). Respective -log10(p-values) are indicated in the last row. Coloured boxes mark genes associated with hemocyte functions or are specifically expressed in *Drosophila* hemocytes (green), genes associated with cell cycle (orange), and genes involved in lipid metabolism (blue).
(TIF)

**S6 Fig. Ush-B activated genes.** Tables of genes that are significantly downregulated (adj. p < 0.05) upon depletion of of Ush-B. Gene symbols are indicated along with the respective fold change relative to cells transfected with control dsRNA (dsEGFP). Respective -log10(p-values) are indicated in the last row. Coloured boxes mark genes associated with hemocyte functions or are specifically expressed in *Drosophila* hemocytes (green), genes associated with cell cycle (orange), and genes involved in lipid metabolism (blue).
(TIF)

**S7 Fig. Cell cycle profiles upon depletion of Ush or NuRD complex components. A** Flow cytometry following PI-staining of S2 cells upon dsRNA-mediated depletion of indicated proteins. dsRNA-transfected cells were fixed, stained with PI and subjected to flow cytometry. Histograms show the number of cells plotted against the PI signal (Area of PE channel). The

diploid cell population (2n) and cells that have undergone replication (4n) are indicated. Transfection of dsEGFP and dsLuc severd as control. Two different dsRNA constructs against Ush (all isoforms) were used (dsUsh #1 & dsUsh #2). **B** Viability assay of S2 cells upon depletion of indicated proteins. Viability of cells transfected with control dsRNA (dsEGFP and dsLuc) or dsRNA constructs targeting Ush (dsUsh #1 and dsUsh #2), Ush-B, dMi-2 and dMTA1-like was measured 96 hours post transfection. Error bars represent the standard deviation from biological triplicates (n = 3) and individual values are indicated with circles.
(TIF)

**S8 Fig.** *Hedgehog* **enhancer activity upon loss of Ush expression.** Lymph glands isolated from larvae that express a dsRNA against Ush in the medullary zone (**A**), or from larvae that carry homozygous Ush mutant alleles (**B**). All larvae carry a construct, reporting the activity of a minimal *Hedgehog* enhancer by GFP expression (hhF4f-GFP; green).
(TIF)

**S1 Table. Occupancy of Ush and dMi-2 at Ush-regulated genes.** Representative Ush-regulated genes of each gene class (investigated in Figs 2E and 6C) are listed. Columns 3 and 4 indicate binding of Ush and dMi-2 to the respective gene loci detected by anti-GFP ChIP sequencing (see Figs 1 and 5).
(PDF)

**S2 Table. Genes deregulated upon Ush-B RNAi.** List of genes that show significant changes (adj. $p < 0.05$) upon depletion of Ush-B. Gene identifiers, fold changes and p-values of each gene are listed. Genes were sorted into the groups "hemocyte-related" (green), "cell cycle" (orange) or "lipid metabolism" (blue) according to the references given in columns 10 and 11.
(PDF)

**S3 Table. Ush and dNuRD regulate lamellocyte differentiation in** *Drosophila* **larvae.** Total numbers of examined larvae and penetrance levels of increased lamellocyte counts associated with Fig 7H. Genotypes and the affected dNuRD complex subunit are listed in columns 1-2.
(PDF)

**S4 Table. Oligonucleotides used in this study.** List of all oligonucleotides and primers used in this study. Sequences and applications are given. References are indicated in column 5.
(PDF)

## Acknowledgments

We thank U.Banerjee, J.Müller, P. Verrijzer, R. Nusse, G. Reuter, C. Wu and J. Mackay for the generous gift of fly lines, antibodies and plasmids, T. Zimmermann for bioinformatical support and Uta-Maria Bauer for critically reading the manuscript.

## Author Contributions

**Conceptualization:** Jonathan Lenz, Olalla Vázquez, Tsuyoshi Tokusumi, Nancy Fossett, Alexander Brehm.

**Data curation:** Andrea Nist, Thorsten Stiewe, Hartmann Raifer.

**Formal analysis:** Jonathan Lenz, Robert Liefke.

**Funding acquisition:** Robert Liefke, Olalla Vázquez, Alexander Brehm.

**Investigation:** Jonathan Lenz, Julianne Funk, Samuel Shoup, Yumiko Tokusumi, Lea Albert, Tsuyoshi Tokusumi.

**Methodology:** Hartmann Raifer, Klaus Förstemann.

**Project administration:** Alexander Brehm.

**Resources:** Robert Schulz, Lea Albert, Klaus Förstemann, Nancy Fossett.

**Supervision:** Alexander Brehm.

**Visualization:** Jonathan Lenz, Robert Liefke, Tsuyoshi Tokusumi, Nancy Fossett.

**Writing – original draft:** Jonathan Lenz, Nancy Fossett, Alexander Brehm.

**Writing – review & editing:** Jonathan Lenz, Nancy Fossett, Alexander Brehm.

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
