## [Decision Letter · Decision Letter 0]

14 Sep 2020

Dear Dr Brehm,

Thank you very much for submitting your Research Article entitled 'Ush regulates hemocyte-specific gene expression, fatty acid metabolism and cell cycle progression and cooperates with dNuRD to orchestrate hematopoiesis' to PLOS Genetics. Your manuscript was fully evaluated at the editorial level and by independent peer reviewers. The reviewers appreciated the attention to an important topic but identified some aspects of the manuscript that should be improved.

We therefore ask you to modify the manuscript according to the review recommendations before we can consider your manuscript for acceptance. Your revisions should address the specific points made by each reviewer. In particular, please note that questions were raised about replicate numbers for some experiments.

[LINK]

Yours sincerely,

Brian Hendrich

Guest Editor

PLOS Genetics

Wendy Bickmore

Section Editor: Epigenetics

PLOS Genetics

Reviewer's Responses to Questions

**Comments to the Authors:**

Reviewer #1: U-shaped, like its mammalian homologue FOG (Friend of GATA) has important role in regulating blood cell development. Previous work indicated that u-shaped expression in precursors, known as prohemocytes, helps to keep them from differentiating. There is evidence to suggest the U-shaped is a repressor but how it, or its homologues, carry out their functions is not currently well understood. The authors have taken advantage of a Drosophila cell line, S2 cells, to identify the U-shaped regulated genes taking two complementary approaches. The first is to generate a GFP-tagged copy of U-shaped and perform ChIP-seq. the second is to deplete U-shaped and determine which RNAs have altered expression. These data reinforce the previously held view that U-shaped/FOG act primarily as repressors.

They also identify a small number of genes that have a modest 2x decrease in expression when U-shaped is depleted, and therefore might be positively regulated by U-shaped. Many of these are cell cycle regulators, and the authors go on to show that u-shaped depletion has an impact on cell cycle progression in the S2 cells. How this relates to in vivo function is less clear as the progenitor population in the lymph gland is a proliferative one.

In considering how U-shaped can positively regulate some genes but repress others, the authors look for interacting factors and, based on previous proteomic data, focus on the NuRD complex. In GST pull-down experiments they map a NuRD -interacting peptide at the N-terminal region of one of the U-shaped isoforms and they show that Mi2 and U-shaped are recruited to some overlapping regions across the genome. Together these data support the model that U-shaped recruits NuRD to repress many of the genes where it is bound. They further demonstrate with one enhancer in vivo, a similar de-repression from depletion of Ush and NuRD complex.

Overall the study is an interesting one that yields new insights into mechanisms of U-shaped function that will be relevant also for the mammalian homologues. The link with NuRD is an exciting one, as are the different roles of U-shaped. Extending more of their findings in vivo would significantly add to the conclusions, and is important to make the claim that their findings show how U-shaped "orchestrates hematopoiesis". Some additional quantifications and clarifications of the data would also strengthen the work.

Specific comments:

1. Given the unexpected results with the cell cycle genes, it would be very valuable to extend to in vivo and test whether the regulation seen in S2 cells is relevant in the pro-hemocytes, by testing some of the targets in a similar way to in Figure 7 and monitoring effects on proliferation.

2. On what basis are the genes in Fig. 2E/6C categorized as hemocyte-related? Two appear to be implicated in immune response but it’s not obvious that others have hemocyte expression/function. It would be of major interest if some of the key differentiation genes were regulated by Ush-Mi2. For this reason, the data for Lozenge (a pro-crystal cell gene) in Supplementary Fig 2 should be included in the main figures. Is this also bound by Mi-2? Does Mi-2/MTA depletion cause derepression? That would add significant weight to the conclusions. Conversely, Msn expression changes in vivo are shown in Fig 7: is there evidence for Ush ± Mi-2 binding to Msn?

3. They find different classes of targets. Given the wealth of data from Drosophila S2 cells, they could compare with existing data sets, chromatin state maps, to see whether the classes are in different chromatin contexts.

4. What approach have they taken for comparing the ChIP seq with the RNA seq data? It is not straightforward to link ChIP peaks to genes for this type of analysis. As S2 cells are not diploid, there is potential for some anomalies in the chromatin data due to variations in ploidy. How have the authors controlled for this in their analysis of the ChIP data?

5. Can they justify using 1bp overlap for the comparison of U-shaped and NuRD binding profiles. If U-shaped is involved in recruiting NuRD the peaks should overlap more significantly (as suggested by the plots). Is the overlap statistically significant? Some of the other S2 ChIP data could be used as comparisons, to demonstrate that the overlaps are specific.

6. Controls with depletion of Ush should be included for the hhF41 enhancer in Figure 7. What are the numbers (N) for the data in Figure 7H.

7. More information should be provided about the CRISPR engineering in the S2 cells. What were the “minor changes” what were the sequences used for targeting and how were the correctly engineered cells selected?

Reviewer #2: Lenz et al further dissect the function of Drospholia Ush in S2 cells and in vivo. They use proteomics, transcriptomics and genetic studies to show that Ush has multiple isoforms, one which interacts with the dNuRD complex and functions differently to the other isoforms. It is a nice study revealing an example of how a single protein can act in multiple mechanisms using alternative splicing.

Main comments:

1. It is currently unclear how many biological replicates were performed: please add this information into all figures, as well as providing an overview of the datasets generated, with the information about repeats and controls into the methods. This is particularly important for the RNAi experiments, where the authors need to show how efficient the knockdown was for each independent replicate, with the resultant overlap of DE genes between replicates.

2. Following on from the previous comment, the RNAi for Ush-b looks less effective than for total Ush (Figure 3D), which might be why you see less genes changing and no cell cycle arrest, rather than differences in isoform function.

3. Please include more information as to how the ChIP peaks are annotated as within Galaxy/Cistrome different regions can be set. How was a promoter peak or an enhancer peak annotated and how was an enhancer peak linked to a gene for comparing the ChIP-seq with DE genes to generate your direct transcriptional targets of Ush.

4. The genes selected in Figure 2E, are these bound by Ush as well as misregulated after dsUsh? If yes, please make that clear in the text, if not Figure 2C-E should be complemented by the same plots but only using direct targets.

a. Likewise, please make clear whether the genes in Figure 6 are bound by Ush and by Mi-2 from your ChIP analyses.

5. Please provide a schematic of the knock-ins of dUsh and dMi-2 including what was knocked in and where in the genes (adding the location of the primers used in Supp Fig1 would be helpful). Plus, it is important to show that these knock-ins did not create a phenotype and alter the normal function of dUsh and dMi-2.

6. S2 cells seem to be the perfect in vitro system for this work as it is a hemocyte-derived cell line. However, how much do they re-capitulate normal fate choices? As the authors mention, changes in Ush are associated with cell fate when pro-hemocytes differentiate to to plasmatocytes, crystal cells and lamellocyte. When you knock-down Ush, and many hemocyte-related genes change, does this result in a change in cell fate for the S2 cells? Likewise, with the knockdown of Mi-2, what is the phenotype in this cell-line?

7. The transcriptomic and proteomic analyses were assessed after 3 or 4 days after RNAi and dsUsh transfected cells do not show normal proliferation after 24 hours. What is the viability of the cells after 3 or 4 days?

8. Please provide details of whether your differentially expressed genes after dsMi-2 is similar to previous reports.

Minor points:

1. As Ush-wt peptide can less efficiently replace GST-mFOG1 than FOG1-wt even though both sequences have the exact same FOG repression motif implies the binding is not completely due to this motif. In addition, the sizes of the Ush peptides seen (180 and 220KDa) do not reflect the size of the predicted proteins (Ush-b is predicted to be 125kDa, Ush-a/c 127kDA, Ush-d 122kDA). This implies there are post-translational modifications, which might play a part in how Ush interacts with NuRD. Therefore, while we agree that the N-terminus of Ush is critical for binding, I would suggest discussing this in the discussion.

2. “However, our findings also suggest that Ush forms complexes with bHLH transcription factors” in the first results section should be toned down. You do not show Co-IPs for Ush and bHLH TFs, but show that they bind to the same locations.

3. A more in depth comparison of what is known about FOG1 and Ush roles in haematopoiesis in the introduction should be included to understand better the potentially conserved roles.

4. In “Ush-B/dNuRD complex does not regulate the cell cycle” results section, “…..depletion of Ush-B, dMi-2 or dMTA1-like did not alter protein expression levels of Cyclin A or Cyclin B (Figure 3C)” should say Figure 3D.

5. Ush hypomorph and null allele names are in very small font and cannot be read in the “Ush/dNuRD regulate hemocyte differentiation in vivo” results section as well as figure legends.

Reviewer #3: In this manuscript, the authors use biochemical and cell biological approaches to examine the function of the Drosophila transcriptional regulator Ush.

The authors first carry out ChIPseq on endogenously tagged Ush in S2 cells. They find it binds to 7K sites that are enriched for promoters – and that the sites are are enriched for GATA sites (and other TF binding sites).

Given that they have conducted a ChIPseq analysis of Ush, the authors should make some comment on whether the protein is likely to bind DNA. In so doing, they could refer to existing data on FOG1.

Next, they use RNAi to deplete Ush and show by RNAseq that 1200 transcripts are upregulated and 600 downregulated. They note that a significant number of these genes are bound by Ush, suggesting that they might be direct Ush targets. Perhaps the authors could indicate in the text what they mean by a gene being bound by USh. Do they mean promoter+gene body? Or something else? Given that we don’t really have very good data on which enhancers are associated with which genes, perhaps their calculated numbers are underestimates, even?

Next they conduct a gene ontology analysis of the genes that were altered in expression under RNAi. Perhaps it would be good if instead or as well they used the set of USh genes that they estimated to be direct targets by ChIPseq. Does this change the ontology analysis?

The authors note that hemocyte-related genes regulated by Ush are overwhelmingly repressed by Ush as are lipid metabolism genes. The opposite is true for cell cycle related genes. This seems an interesting finding.

RT-qPCR was carried out to corroborate the RNAseq data. The effect sizes are large for the hemocyte- and lipid metabolism-related genes. However, the effect sizes are quite small for the cell cycle genes – all less than two-fold (though the authors do show an effect on cell cycle subsequently). The authors mark these results as statistically significant. I have a couple of issues with this presentation. One is that the figure legend says that the error bars represent the SD from technical triplicates. The authors should be aware that technical replicates cannot be used for drawing statistical inferences. Either they have just written the wrong thing in the legend (and their replicates are *biological* replicates) or they need to remove all the error bars shown and remove the T test calculations. Second, even if their data really do represent triplicate independent measurements (and don’t get me started about the independence of replicates in most cell biology experiments…), the less-than-two-fold effect sizes for the cell cycle genes would lead me to conclude that it’s quite unlikely that there is a real/interesting effect here (based on these data alone). For one thing, I would have thought it’s quite hard, given the nature of a qPCR experiment, to be confident about a two-fold effect – the experiment involves measuring midpoints that arise from PCR cycles (doublings), so it feels like you’re pushing the limits of the technique to try to draw inferences about two-fold (or smaller) effects, unless you have true biological replicates (which cell biology experiments often don’t – different aliquots of the same cells from the freezer etc). As a corollary, the authors should note that there is a move in the literature to showing actual data points on bar graphs rather than just the bars. In this case, if their data points are technical replicates, then they only have one data point to show for each bar (the mean of the technical triplicates). Finally, these comments also apply to Figure 6 (in particular Fig 6C, which talks about technical triplicates). Really, the world would be a better place in most cases without these t-tests – given how unstable p values are for small sample sizes and small effect sizes (and therefore how little confidence can be attached to them) – and how arbitrary the 0.05 cutoff is, but that’s a battle for another day.

That was a bit of a rave, but I think there is *something* in there for the authors to address… and either way it doesn’t significantly affect the main thrust of the manuscript.

Flying in the face of the questions I have about the qPCR data, the authors go on to show an effect of Ush knockdown on cell cycle.

The authors next examine the splice isoforms of Ush, to ask whether activation vs repression functions of Ush might be associated with these isoforms. Interestingly, they show that only one of the isoforms (UshB) contains an N-terminal sequence that is already known from published work on FOG1 to interact with the NuRD complex. They nicely demonstrate that this is the case for Ush – only UshB interacts with NuRD components and the interaction is clearly mediated by the N-terminal motif identified previously in other proteins such as FOG1.

I would just note in this section that the authors say things such as “We conclude that dMi-2 specifically interacts with the Ush-B isoform”. However, the nature of a coIP is that you cannot identify direct interactions – only conclude that two proteins can exist in the same complex. Many people would read “specifically interacts” as meaning a direct interaction, which is a bit misleading I think (unintentionally here, I have no doubt).

The authors show that dMi2 and Ush ChIPseq datasets show a lot of peak overlap. It would be nice to see a breakdown of the overlapping vs non-overlapping peaks. Are they enriched/depleted for active/repressed genes compared to all dMi2/Ush peaks? What about promoters/enhancers etc? GATA sites? It feels like there is information in here that could be teased out.

It would also be nice to know what is meant by the peaks ‘overlapping’. That would involve some assessment of peak width in the experiments. For example, if the peaks are 500 bp wide (I just made that number up), does overlap mean overlap of 20% of the peak area? 50%? Something else? Knowing that dMi2 and Ush were 500 bp apart is different from knowing they’re 20 bp apart on DNA.

Next, the authors carry out RNAseq and RT-qPCR on S2 cells specifically depleted for UshB (and it’s very nice that they are able to do this) or for dMi2. They show that these treatments to not alter expression of cell cycle genes or (nearly all) lipid metabolism genes. They comment that genes involved in hemocyte development *are* affected, but they are a bit vague: “changes in several genes related to…”. Could they be more quantitative here please? And it would be good if they compared the genes that change in the UshB knockdown with those that change in the Ush knockdown. What is the overlap? In each functional class?

They go on to show that NuRD and Ush are involved in specific hematopoietic events. I won’t try to comment on the fly biology experiments as I am not an expert in this area.

In their discussion, the authors comment on the Tal1/LMO2/LDB1 (they need to correct the “LBD1” that they write) complex in mammals. It would be good for them to comment on the biochemistry of the fly versions of these proteins: namely Chip, LMO, DA/AC/SC. I thought there was evidence that these proteins come together with Pannier to form complexes – which makes it very likely that Ush is involved too. Are there ChIPseq data on these other proteins that could be compared to the authors’ Ush ChIPseq data?

The authors also comment on the FOG repression motif and say that they’ve identified one in OAZ. It’s worth noting that this motif doesn’t have to be at the N-terminus of a protein to bind to NuRD (e.g., PHF6).

The really interesting question that this work raises – that the authors try to address in their discussion, but can’t really – is why UshB depletion only affects a functionally distinct set of Ush target genes out of the sets identified in their earlier Ush knockdown experiment (that is, the hemocyte ones). They suggest that the hemocyte genes might be particularly dependent on NuRD or high levels of Ush. This is possible but it’s hard at this stage to know why that might be. That’s OK – they can’t answer *every* question in this paper.

In this regard, it would have been nice if the authors could have determined which sites are bound by UshB in the genome and which by non-NuRD binding isoforms. Might this not give some insight into whether UshB is distributed across all Ush sites or enriched at a subset. They *could* have done ChIPseq on their UshB-depleted cells to answer this question, couldn’t they? I don’t expect them to do that experiment in COVID world, but they could at least comment on it one way or the other perhaps.

Overall, this manuscript is of high quality, well written and the story presented in a logical and straightforward way. The authors dissect Ush function in a clear and rigorous way, and are able to present several interesting conclusions that are of relevance to researchers trying to understand the mechanisms underlying gene regulation in eukaryotic organisms. Thus, I am of the view that this work is appropriate for publication in PLoS Genetics.

Minor points

In the Introduction, the authors state:

“Changes in Ush levels govern cell fate choice: The stem cell-like pro-hemocytes express high levels of Ush. Ush expression is downregulated to lower levels as pro-hemocytes differentiate into plasmatocytes and crystal cells and completely shut off during lamellocyte differentiation (Fossett, 2013).”

However, their statement that Ush governs cell fate choice can’t be inferred from the following statements – one can only infer that Ush expression and hemocyte type are correlated. If it is known that the relationship is causal, ,the authors should instead mention data that make this connection.

“Established roll” should be “Established role”

**Have all data underlying the figures and results presented in the manuscript been provided?**

Reviewer #1: Yes

Reviewer #2: Yes

Reviewer #3: Yes

PLOS authors have the option to publish the peer review history of their article (what does this mean?). If published, this will include your full peer review and any attached files.

Reviewer #1: No

Reviewer #2: **Yes: **Anzy Miller

Reviewer #3: No

---

## [Decision Letter · Decision Letter 1]

5 Dec 2020

Dear Dr Brehm,

Thank you very much for submitting your Research Article entitled 'Ush regulates hemocyte-specific gene expression, fatty acid metabolism and cell cycle progression and cooperates with dNuRD to orchestrate hematopoiesis' to PLOS Genetics.

The manuscript was fully evaluated at the editorial level and by independent peer reviewers. The reviewers are mostly satisfied with the revised draft but still have a some minor issues which need to be addressed.

We therefore ask you to modify the manuscript according to the review recommendations. Your revisions should address the specific points made by the two reviewers who suggest edits.

[LINK]

Yours sincerely,

Brian Hendrich

Guest Editor

PLOS Genetics

Wendy Bickmore

Section Editor: Epigenetics

PLOS Genetics

Reviewer's Responses to Questions

**Comments to the Authors:**

Reviewer #1: The authors have performed additional analysis to clarify their results and have addressed many of the concerns raised. There remain some aspects that require further clarification.

First, to what extent are the S2 data representative of Ush role in the lymphgland. There are mixed messages about this. E.g. the answer to Reviewer 1 point 1 seems to suggest that depleting Ush has converse effects on proliferation in vivo. The discussion may be misleading on this point. And they need to be cautious about how much of the gene programmes can be extrapolated.

Second, 2 concerns about the ChIP data. (i) There has only been one replicate performed. (ii) There is no GFP control - it’s possible that the ChIPs are detecting “open chromatin” that would overlap between the two samples. To some extent the concern re (i) is mitigated by the similarities to the existing Mi-2 RNA data. They could use the publicly available Mi-2 ChIP data from S2 cells and/or some of the other GFP ChIPs from S2 cells to discard these concerns.

Third, what is the significance of the interaction between Ush and NuRD? In the rebuttal (Reviewer 1 Point 5) they say “Therefore, even if Ush was actively recruiting NuRD to chromatin (which we do not claim)” If Ush is not recruiting NuRD, what is the role of the direct interactions they have detected? This is a major element of the paper so they need to make some explanation about the significance of these interactions. Indeed in the abstract they refer to “the Ush/NuRD complex”. What does Ush contribute if not helping to direct NuRD to specific targets?

Minor Points:

Reviewer 2 point 4:

The table is a valuable addition to the paper and helps the understanding, I suggest it is included.

The discussion is very long and could be significantly condensed.

Reviewer #2: The authors have addressed all my comments well.

However, I was surprised to find there was only one replicate of the ChIP-seq experiments performed. While the authors compare the Mi-2 ChIP with two other data sets, the ChIP-seq for Ush is not- presumably because there are no published data sets already. There is high variability between ChIP experiments, and as such repeats would be beneficial to generate high confidence binding sites. While I am not asking for repeats to be performed, I would ask that a sentence be included somewhere in the manuscript to alert the reader that these sites maybe not be a comprehensive list of Ush binding sites due to only have one repeat.

Reviewer #3: The authors have done an excellent job of addressing my comments (and those of the other reviewers I think). I am fully supportive of this manuscript being accepted for publication.

**Have all data underlying the figures and results presented in the manuscript been provided?**

Reviewer #1: Yes

Reviewer #2: Yes

Reviewer #3: Yes

PLOS authors have the option to publish the peer review history of their article (what does this mean?). If published, this will include your full peer review and any attached files.

Reviewer #1: No

Reviewer #2: **Yes: **Anzy Miller

Reviewer #3: **Yes: **Joel Mackay

---

## [Editor Report · Decision Letter 2]

20 Dec 2020

Dear Dr Brehm,

We are pleased to inform you that your manuscript entitled "Ush regulates hemocyte-specific gene expression, fatty acid metabolism and cell cycle progression and cooperates with dNuRD to orchestrate hematopoiesis" has been editorially accepted for publication in PLOS Genetics. Congratulations!

Yours sincerely,

Brian Hendrich

Guest Editor

PLOS Genetics

Wendy Bickmore

Section Editor: Epigenetics

PLOS Genetics

Comments from the reviewers (if applicable):

**Data Deposition**

http://datadryad.org/submit?journalID=pgenetics&manu=PGENETICS-D-20-01249R2

**Press Queries**

---

## [Editor Report · Acceptance letter]

28 Jan 2021

PGENETICS-D-20-01249R2 

Ush regulates hemocyte-specific gene expression, fatty acid metabolism and cell cycle progression and cooperates with dNuRD to orchestrate hematopoiesis 

Dear Dr Brehm, 

We are pleased to inform you that your manuscript entitled "Ush regulates hemocyte-specific gene expression, fatty acid metabolism and cell cycle progression and cooperates with dNuRD to orchestrate hematopoiesis" has been formally accepted for publication in PLOS Genetics! Your manuscript is now with our production department and you will be notified of the publication date in due course.

With kind regards,

Alice Ellingham

PLOS Genetics

On behalf of:
